# Calcium signaling regulates apoptosis-induced proliferation in *Drosophila*

Komal Panchal Suthar[1], Caitlin Hounsell[2], Yun Fan[2], Andreas Bergmann[1]*

**1** Department of Molecular, Cell and Cancer Biology, UMass Chan Medical School, Worcester, Massachusetts, United States of America, **2** The University of Birmingham, Edgbaston, Birmingham, United Kingdom

* Andreas.bergmann@umassmed.edu

## Abstract

Caspases, traditionally viewed as mediators of apoptosis and tumor suppressors, have also been shown to promote cell proliferation and to contribute to tumor growth. For example, the initiator caspase Dronc (the *Drosophila* orthologue of Caspase-9) can trigger apoptosis-induced proliferation (AiP), a process where apoptotic cells generate mitogenic signals for compensatory proliferation independently of their apoptotic function. AiP is crucial for homeostatic cell turnover, wound healing, and tissue regeneration. Previously, we established that Dronc activates the NADPH oxidase DUOX at the plasma membrane, resulting in the production of extracellular reactive oxygen species (ROS) which are required for AiP. However, the mechanism by which Dronc activates DUOX has remained elusive. Here, we identified Dronc-dependent $Ca^{2+}$ entry into the cytosol as a significant factor for DUOX activation and AiP. Three cell surface $Ca^{2+}$ channels of the TRP family mediate $Ca^{2+}$ influx in a non-redundant fashion. Additionally, calcium-induced calcium release (CICR) from the ER was identified as another source of cytosolic $Ca^{2+}$ during AiP. Notably, DUOX itself acts as a $Ca^{2+}$ effector in AiP, requiring $Ca^{2+}$ binding for its activation. These findings highlight the importance of $Ca^{2+}$ signaling in AiP and provide insights into how similar signaling mechanisms might operate in vertebrates.

## Introduction

Apoptosis is a physiological form of cell death that accounts for most cell death in eukaryotes [1,2] and is essential for normal development, homeostasis, and immune defence by elimination of damaged, unnecessary, or potentially harmful cells from an organism [3]. Alterations of apoptosis can result in various diseases. Apoptotic resistance can lead to uncontrolled cell proliferation such as cancer [4,5] and auto-immune diseases [6], while exaggerated apoptosis contributes to neurodegenerative disorders and immunodeficiency [7]

**Data availability statement:** All relevant data are within the paper and its Supporting information files.

**Funding:** This work was supported by a grant from the National Institute of General Medical Sciences (NIGMS) under award number R35GM118330 to AB, and the UK Research and Innovation (UKRI) BBSRC under award number BB/Z516971/1 to YF. The funders had no role in study design, data collection and analysis, decision to publish, or preparation of the manuscript. https://www.nigms.nih.gov/ https://www.ukri.org/councils/bbsrc/.

**Competing interests:** The authors have declared that no competing interests exist.

**Abbreviations:** AiP, apoptosis-induced proliferation; CICR, calcium-induced calcium release; DHE, dihydroethidium; Dpp, Decapentaplegic; EGF, epidermal growth factor; ER, endoplasmic reticulum; IAPs, inhibitor of apoptosis proteins; RNAi, RNA interference; ROS, reactive oxygen species; RyR, Ryanodine Receptor; TRP, transient receptor potential; Wg, Wingless.

A specific class of Cys proteases, termed Caspases, are the main executioners of apoptosis which cleave key intracellular substrates to induce cell death [8–10]. There are two types of caspases: initiator caspases such as Caspase-9 and effector (or executioner) caspases such as Caspase-3 and Caspase-7. Initiator caspases play a crucial role in the early stages of apoptosis as they are responsible for activating downstream effector caspases, which carry out the apoptotic cell death process by cleaving a large number of intracellular substrates [3,5,8,11].

The fruit fly, *Drosophila melanogaster*, has emerged as a powerful model organism for studying apoptosis [3,12]. In *Drosophila*, apoptosis is orchestrated by homologs of mammalian caspases. Of the seven caspases in *Drosophila*, only Dronc, a caspase-9-like initiator caspase, and DrICE and Dcp-1, caspase-3-like effector caspases, are critical for apoptosis [12,13]. Dronc cleaves and activates DrICE and Dcp-1, which perform the final dismantling of the cell [12,13]. Furthermore, the inhibitor of apoptosis proteins (IAPs), most notably *Drosophila* IAP1 (DIAP1), and their antagonists, Reaper, Hid, Grim, Sickle, and Jafrac-2 regulate apoptosis in *Drosophila* by modulating caspase activity [14–17]. These genes ensure that apoptosis occurs only under appropriate conditions to maintain cellular homeostasis.

However, while Caspases are best-known for their role in apoptosis, they also perform several non-apoptotic functions that are crucial for normal cellular processes, for example, caspase-mediated signaling required for tissue regeneration through a process called apoptosis-induced proliferation (AiP) [9,18–21]. AiP is a particular subtype of compensatory proliferation in which apoptotic cells, instead of being passively removed, actively signal to neighboring cells to proliferate [19,22–26]. This process plays a significant role in tissue regeneration, wound healing, and cancer development [19,27,28], demonstrating that apoptosis is not solely a mechanism for cellular death but also a trigger for compensatory growth. The initiator caspase Dronc plays a key role in AiP by promoting the release of mitogens, such as Wingless (Wg), Epidermal growth factor (EGF), and Decapentaplegic (Dpp), which stimulate cell proliferation in surviving neighboring cells [29–34].

A classic model of examining AiP in *Drosophila* uses the expression of the baculoviral protein p35, which acts as a potent inhibitor of the effector caspases Drice and Dcp-1, but not of the initiator caspase Dronc [35–37]. If *p35* expression is combined with expression of the IAP antagonist *hid*, "undead" cells are produced, which are immortalized and continue to produce signals for the proliferation of neighboring cells, resulting in overgrowth of the undead tissue [34]. Thus, undead cells are key experimental tools for studying AiP, as they uncouple the apoptotic and proliferative functions of the initiator caspase Dronc. In *Drosophila*, AiP has been studied extensively in imaginal discs—larval epithelial tissues that give rise to adult structures during metamorphosis such as eyes, wings, legs, etc. If undead cells are created in the eye antennal imaginal discs using *ey-Gal4* (denoted as *ey>hid,p35*), adult flies are recovered, which display a range of overgrown heads (Fig 1A) [34,38].

Genetic screening in *Drosophila* for suppressors of *ey>hid,p35*-induced overgrowth identified several genes that regulate AiP. This work revealed a number of key signaling events for AiP to occur. First, Dronc is transported to the basal side of the

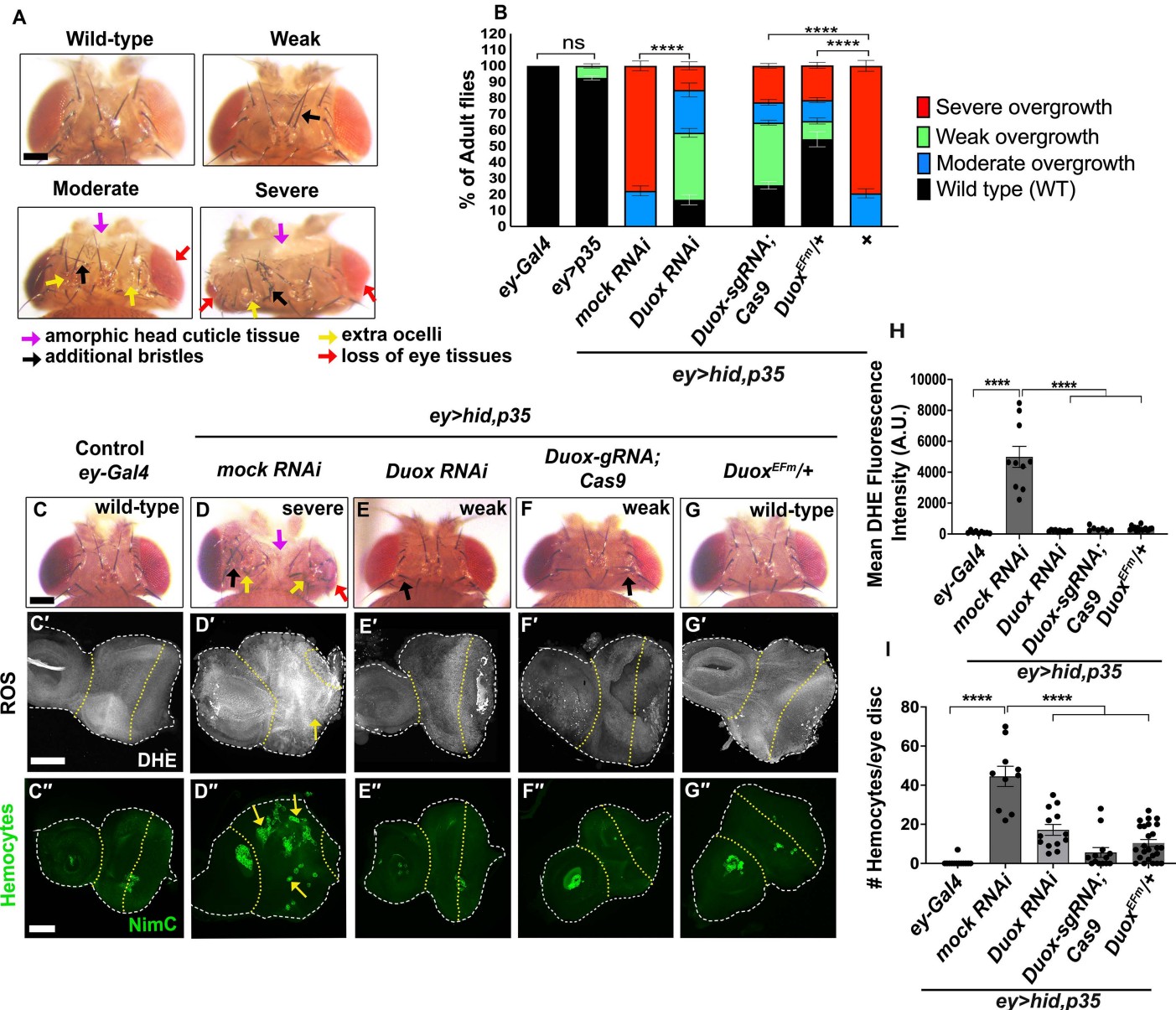

**Fig 1. Loss of EF-hand function of DUOX suppresses undead overgrowth and all AiP markers. (See also** S1 **and** S2 Figs**).** Yellow dotted lines highlight the *ey-Gal4*-expressing area of the eye discs. The disc outline is marked by white dashed lines. Scale bars: 100 μm (A, C–G), 50 μm (C′–G′, C″-G″). **(A)** Representative examples of adult fly heads, depicting either a wild-type head or a range of overgrown phenotypes categorized from weak to moderate to severe. The severe phenotype is characterized by overgrowth of the head, including amorphic head capsule tissue (purple arrows), numerous additional bristles (up to 15) (black arrows), duplicated ocelli (yellow arrows), and often a reduction of the eye size (due to the expansion of the head capsule area into the posterior eye field during development) (red arrows). Moderate overgrowth is characterized by head enlargement with fewer extra bristles (up to 6) (black arrow), duplicated ocelli (yellow arrows), and smaller eye size (red arrow). Weak overgrowth is characterized by only mild head enlargement with one to two additional bristles (black arrow). **(B)** Quantification of the suppression of head overgrowth of adult *ey>hid,p35* flies across the indicated genetic backgrounds. Flies were scored as wild type (wt) (black bars), weak (green bars), moderate (blue bars) or severe overgrowth (red bars) according to the classification in Fig 1A. *n* = 100 flies counted per genotype in three independent experiments. **(C)** Wild-type fly with normal head morphology. **(D)** Severely overgrown head of *ey>hid,p35* flies expressing mock (*Luciferase)* RNAi. Arrows point to amorphic tissues, additional bristles and ocelli, and reduced eye tissue. **(E, F)** Representative examples of suppressed overgrowth of *ey>hid,p35* flies expressing *UAS-Duox* RNAi **(E)**, or *Duox-gRNA;UAS-Cas9.P2* **(F)**. Arrows point to one or two extra bristles indicating mild overgrowth. **(G)** Representative examples of completely suppressed overgrowth of *ey>hid,p35* flies in the background of *DuoxEFm/+*. **(C′–G′)** Confocal images showing third instar larval eye imaginal discs of control (*ey-Gal4*) **(C′)**, and *ey>hid,p35* discs expressing mock (*Luciferase)* RNAi **(D′)**, *UAS-Duox* RNAi **(E′)**, *Duox-gRNA;UAS-Cas9.P2* **(F′)**, and

*Duox^{EFm}/+* (H′) labeled for ROS with dihydroethidium (DHE) dye. The yellow arrow in (I) indicates DHE-positive cells. **(C″–G″)** Confocal images showing hemocyte labeling using the plasmatocyte-specific anti-NimC antibody in eye imaginal discs of control (*ey-Gal4*) (C″), and *ey>hid,p35* discs expressing mock (*Luciferase*) RNAi (D″), *UAS-Duox* RNAi (E″), *Duox-gRNA;UAS-Cas9.P2* (F″), and *Duox^{EFm}/+* (H″). In *ey-Gal4* control discs, hemocytes are present as cell aggregates in the antennal portion of the disc and posterior to the morphogenetic furrow at the eye disc (C″). At undead *ey>hid,p35* discs expressing mock (*luciferase*) RNAi, an increased number of hemocytes is present in overgrown areas (yellow arrows) (D″). This increased number of the hemocytes at undead eye discs is strongly suppressed by expressing *UAS-Duox* RNAi, *Duox-gRNA;UAS-Cas9.P2* and *Duox^{EFm}/+* (E″–G″). **(H)** Quantification of DHE fluorescence in (C′–G′). Data from $n = 9$ (*ey-Gal4*), 10 (*mock* RNAi), 11 (*UAS-Duox* RNAi), 7 (*Duox-gRNA;UAS-Cas9*), and 16 (*Duox^{EFm}/+*) discs were analyzed in three independent experiments. A.U.—arbitrary units. **(I)** Quantification of the number of hemocytes per disc in (C″-G″). Data from $n = 13$ (*ey-Gal4*), 10 (*mock* RNAi), 13 (*UAS-Duox* RNAi), 13 (*Duox-gRNA;UAS-Cas9*), and 25 (*Duox^{EFm}/+*) discs were analyzed in three independent experiments. Levels of significance are depicted by asterisks in the figures: ****$p < 0.0001$. The data underlying the graphs shown in this figure can be found in S1 Data.

plasma membrane by Myo1D, an unconventional class 1 myosin, and LimK1, a stabilizer of actin filaments [39,40]. At the membrane, Dronc either directly or indirectly activates the NADPH oxidase DUOX, leading to the production of extracellular reactive oxygen species (ROS) [38]. ROS attract and activate macrophage-like immune cells termed hemocytes which release signaling factors, including the TNF homolog Eiger to stimulate JNK signaling in undead cells [38,41,42]. JNK then drives the expression of endogenous *hid* and *reaper* genes, creating an amplification loop that sustains AiP signaling in undead cells [38,43]. Additionally, JNK induces expression of the mitogenic factors Wg, EGF, and Dpp, which promote the proliferation of surviving neighboring cells [29,30,44].

Calcium ions (Ca²⁺) function as versatile intracellular secondary messengers in a wide range of physiological processes [45–49]. While Ca²⁺ signaling has been best studied in excitable cells such as muscles and neurons, it is also essential in non-excitable cells such as epithelial cells [50–57] where it controls numerous physiological processes such as cell proliferation, differentiation, cellular migration, barrier function, secretion, immune responses, wound healing, and apoptosis [58–62]. The range of Ca²⁺ signaling in epithelial cells and its role in the regulation of epithelial characteristics are poorly understood.

Cytosolic Ca²⁺ levels are tightly regulated by Ca²⁺ channels, pumps, and exchangers located in the plasma membrane and intracellular organelles such as the endoplasmic reticulum (ER) and mitochondria [63,64]. Upon receiving external or internal stimuli, Ca²⁺ can be released into the cytosol, where it binds to target proteins and activates effector pathways [64]. Often, there is interplay between extracellular and intracellular Ca²⁺ sources to ensure precise control over cytosolic Ca²⁺ levels. For example, calcium-induced calcium release (CICR) from the ER amplifies the initial elevations of intracellular Ca²⁺ levels from the plasma membrane [65,66], and contributes to flashes and waves of Ca²⁺ activity [67]. Ca²⁺ flashes are a distinctive feature of epithelial Ca²⁺ signaling and are involved in processes such as gene expression, proliferation, and wound healing where Ca²⁺ waves propagate across epithelial layers to coordinate cell migration and tissue repair [57,59,68]. These flashes result from the interplay between Ca²⁺ influx, release, and re-uptake mechanisms that generate temporal Ca²⁺ patterns [69,70].

A key unresolved question in AiP research is the mechanism of DUOX activation in undead cells. DUOX contains two canonical Ca²⁺ binding EF-hand motifs on an intracellular loop, where cytosolic Ca²⁺ binding is essential for DUOX activation in both mammals and *Drosophila* [68,71–73]. For example, during embryonic wound repair in *Drosophila*, Ca²⁺ influx mediates activation of DUOX [68]. Furthermore, upon bacterial infection in the *Drosophila* intestine, Ca²⁺ binding activates DUOX for ROS generation, which serves as an antibacterial response [74,75]. Moreover, in zebrafish, tissue injury triggers early recruitment of leukocytes to the wound and it also induces an inflammatory response through activation of the NF-κB signaling pathway via activation of DUOX1, which induces the production of $H_2O_2$ and modulates the in vivo inflammatory response [76].

Here, we investigated the role of Ca²⁺ signaling for DUOX activation in AiP regulation. Our studies revealed that mutation of the EF-hand motifs of DUOX abolished ROS production and blocked AiP. Consistent with these findings, we detected Dronc-dependent cytosolic Ca²⁺ signaling in undead cells. Rather than maintaining steady-state levels, Ca²⁺ signaling in undead tissue occurred as flashes. We identified two distinct sources of Ca²⁺ influx: extracellular Ca²⁺

entry through three TRP family Ca²⁺ channels in a non-redundant manner, and CICR from the ER through the Ryanodine Receptor (RyR). The non-redundant roles of TRP channels in Ca²⁺ influx, RyR-mediated CICR, and DUOX's Ca²⁺-dependent ROS production exemplify a complex yet robust system that enables cells to respond effectively to damage and stress. These findings provide valuable insights into conserved Ca²⁺ signaling pathways and their potential as therapeutic targets in diseases involving impaired tissue repair or dysregulated proliferation.

## Results

### A *Duox* mutant unable to bind Ca²⁺ suppresses all AiP phenotypes

Co-expression of *hid* and *p35* under *ey-Gal4* control (*ey>hid,p35*) generates a range of AiP-induced overgrowth phenotypes in the undead head capsule, categorized as weak, moderate, or severe (Fig 1A). Quantitative analysis reveals that the majority of *ey>hid,p35* flies (79%) display severe overgrowth, while the remaining 21% show moderate overgrowth phenotypes (Fig 1B).

Previously, we demonstrated that *Duox* knockdown by RNA interference (RNAi) moderately suppresses the AiP-induced overgrowth phenotype of *ey>hid,p35* animals [38] (Fig 1B–1E). To further validate DUOX's requirement for AiP, we performed tissue-specific CRISPR/Cas9-mediated targeting of *Duox* in eye-antennal discs of *ey>hid,p35* animals, expressing *UAS-Cas9* under *ey-Gal4* control. This approach similarly resulted in moderately strong suppression of *ey>hid,p35*-induced overgrowth (Fig 1B and 1F).

To specifically examine the role of Ca²⁺ binding in DUOX regulation, we targeted two highly conserved glutamate residues, Glu879 and Glu915, which occupy the canonical "Z" position within adjacent EF-hand motifs. In EF-hand Ca²⁺-binding loops, this terminal glutamate provides critical bidentate coordination of Ca²⁺, and substitution with glutamine is known to markedly reduce or abolish Ca²⁺ affinity without disrupting overall protein folding [77,78]. Guided by this precedent, we replaced Glu879 and Glu915 with glutamine (E879Q, E915Q), thereby generating a mutant allele of endogenous *Duox*, referred to as *Duox^EFm*, that is predicted to selectively disrupt Ca²⁺ chelation while preserving protein integrity. Heterozygously, *Duox^EFm*/+ suppressed *ey>hid,p35*-induced overgrowth to a similar extent as *Duox* RNAi or CRISPR/Cas9-mediated targeting of *Duox* (Fig 1B and 1G). The ability of *Duox^EFm*/+ to dominantly suppress the undead overgrowth of *ey>hid,p35* animals indicates that intact EF-hand motifs, and thus Ca²⁺-dependent activation of DUOX, are essential for its function in undead AiP signaling.

Consistently, *Duox^EFm*/+ suppressed multiple AiP markers in *ey>hid,p35* background. First, ROS production, the primary function of DUOX and a characteristic marker of undead signaling [38], was reduced to levels comparable to *Duox* RNAi treatment (Fig 1C′–1G′; quantified in Fig 1H). Second, hemocyte recruitment was similarly decreased, matching the reduction seen with *Duox* RNAi (Fig 1C″–1G″; quantified in Fig 1I). Finally, similar to expression of *Duox* RNAi, *Duox^EFm*/+ in the *ey>hid,p35* background suppressed both JNK pathway activation, as measured by the marker MMP1, and Wg expression (S1 Fig).

To further probe the requirement of the EF-hand motifs, we expressed a *Duox* transgene that lacks these motifs (*UAS-Duox^ΔEF*) in *ey>hid,p35* background. Like *Duox^EFm,* expression of *UAS-Duox^ΔEF* suppressed *ey>hid,p35*-induced overgrowth (S2A and S2B Fig), suggesting that it acts as a dominant-negative mutant. Similarly, expression of *UAS-Duox^ΔEF* reduced ROS production, decreased hemocyte recruitment, and suppressed both JNK activation (MMP1) and Wg expression (S2 Fig).

Together, these results demonstrate that the EF-hand motifs of DUOX are essential for both ROS generation and for subsequent ROS-dependent events in undead AiP signaling.

### Release of Ca²⁺ during AiP in a Dronc-dependent manner

Because the only known function of EF-hand motifs is Ca²⁺ binding [72,73], we investigated the involvement of Ca²⁺ in AiP signaling using the cytosolic Ca²⁺ reporter GCaMP6s [79]. Control eye imaginal discs (*ey-Gal4* and *ey>p35*) showed very

low GCaMP6s activity anterior to the morphogenetic furrow where *ey-Gal4* is expressed (Fig 2A and 2B; quantified in Fig 2D). In contrast, undead (*ey>hid,p35*) eye imaginal discs displayed very strong GCaMP6s activity in this region (Fig 2C, white arrows; quantified in Fig 2D), indicating significantly increased Ca²⁺ signaling. Time-lapse imaging revealed highly dynamic GCaMP6s activity in undead discs (S1–S3 Movies). Moreover, we detected irregular Ca²⁺ flashes in undead discs that were absent in controls (Fig 2E–2G; quantified in Fig 2H and S1–S3 Movies). These flashes varied between discs, persisted for seconds (Fig 2G and S3 Movie; for complete individual recordings of the Ca²⁺ traces see S3 Fig), and were restricted to the undead (*ey-Gal4*-expressing) domain. Within this domain, however, the flashes occurred without an obvious spatial pattern. Together, these data demonstrate that Ca²⁺ signaling in undead (*ey>hid,p35*) eye imaginal discs occurs as dynamic oscillations.

Importantly, Ca²⁺ signaling in undead tissue depends on the initiator caspase Dronc. RNAi targeting *Dronc* suppressed both the overall GCaMP6s fluorescence of undead (*ey>hid,p35*) discs (Fig 2I and 2J; quantified in Fig 2K) and the Ca²⁺ flashes of undead (*ey>hid,p35*) discs (Fig 2L and 2M; quantified in Fig 2N and S4–S5 Movies and S3 Fig). These data demonstrate that Ca²⁺ release and its flashes are direct consequences of *hid,p35*-induced undead signaling.

### Identification and characterization of Ca²⁺ transporters involved in AiP

Next, we sought to identify the Ca²⁺ channel(s) that mediate cytosolic Ca²⁺ influx. Since Dronc localizes to the plasma membrane in undead cells [39,80], where it might directly or indirectly control Ca²⁺ influx, we focused on channels that transport Ca²⁺ across the plasma membrane. Transient receptor potential (TRP) channels represent one major class of plasma membrane Ca²⁺ transporters that respond to various extra- and intracellular signals potentially generated by undead cells [81,82]. Among the 13 TRP channels encoded in the *D. melanogaster* genome, RNAi screening identified three of them (*TrpM*, *TrpA1*, *Pkd2*) as moderately strong suppressors of undead overgrowth (Figs 3A, 3B, and S4A). Their suppression levels matched those observed with *Duox* inactivation or the *Duox^EFm* mutant (compare Fig 1B to Fig 3B). We validated these findings using CRISPR/Cas9-mediated gene inactivation for *TrpM* and *TrpA1,* and additionally confirmed the role of *TrpA1* using a null mutant allele (*TrpA1^ins*) (S5A and S5B Fig).

Consistent with the function of these TRP channels as Ca²⁺ transporters, inactivation of *TrpM, TrpA1*, and *Pkd2* in undead (*ey>hid,p35*) background blocked the GCaMP6s signal in eye imaginal discs (Fig 3C–3F; quantified in Fig 3S). Notably, inactivating any single channel alone strongly abolishes GCaMP6s signaling, suggesting that these channels operate in an interdependent non-redundant manner. Furthermore, RNAi targeting any of these three TRP channels effectively suppressed the Ca²⁺ flashes typically observed in undead discs (Fig 3G–3J; quantified in Fig 3T and S4 and S6–S8 Movies; for complete individual recordings of the Ca²⁺ traces see S6 Fig).

A key potential function of Ca²⁺ signaling in undead cells is the activation of DUOX through binding to the EF-hand motifs, leading to ROS generation. Using dihydroethidium (DHE) as a ROS indicator, we found that RNAi of either *TrpM, TrpA1*, or *Pkd2* channels significantly impacted ROS generation in undead (*ey>hid,p35*) imaginal discs (Fig 3K–3N; quantified in Fig 3U). Consistently also, given that DUOX-generated ROS are essential for hemocyte recruitment to undead discs [38,41], inactivation of either *TrpM, TrpA1*, or *Pkd2* substantially prevented the recruitment of hemocytes to the undead disc (Fig 3O–3R; quantified in Fig 3V). Furthermore, knockdown of *TrpM, TrpA1*, and *Pkd2* in undead cells significantly suppressed both JNK activity and Wg expression (S4B–S4K Fig).

Additionally, the *TrpA1* null mutant allele, *TrpA1^ins,* dominantly suppressed the GCaMP6s signal, and Ca²⁺ flashes in undead (*ey>hid,p35*) eye imaginal discs (S5C–S5H and S6F–S6G Figs and S4 and S10 Movies). Consistently, *TrpA1^ins*/+ suppressed all AiP markers including ROS levels, hemocyte recruitment, JNK activity, and Wg expression in undead eye imaginal discs (S5I–S5T Fig).

These findings demonstrate that disrupting Ca²⁺ signaling produces phenotypes identical to those observed with *Duox^EFm* and *UAS-Duox^ΔEF* expression in undead discs, suggesting that a primary function of Ca²⁺ is to activate DUOX through binding to its EF-hand motifs, thereby enabling ROS generation.

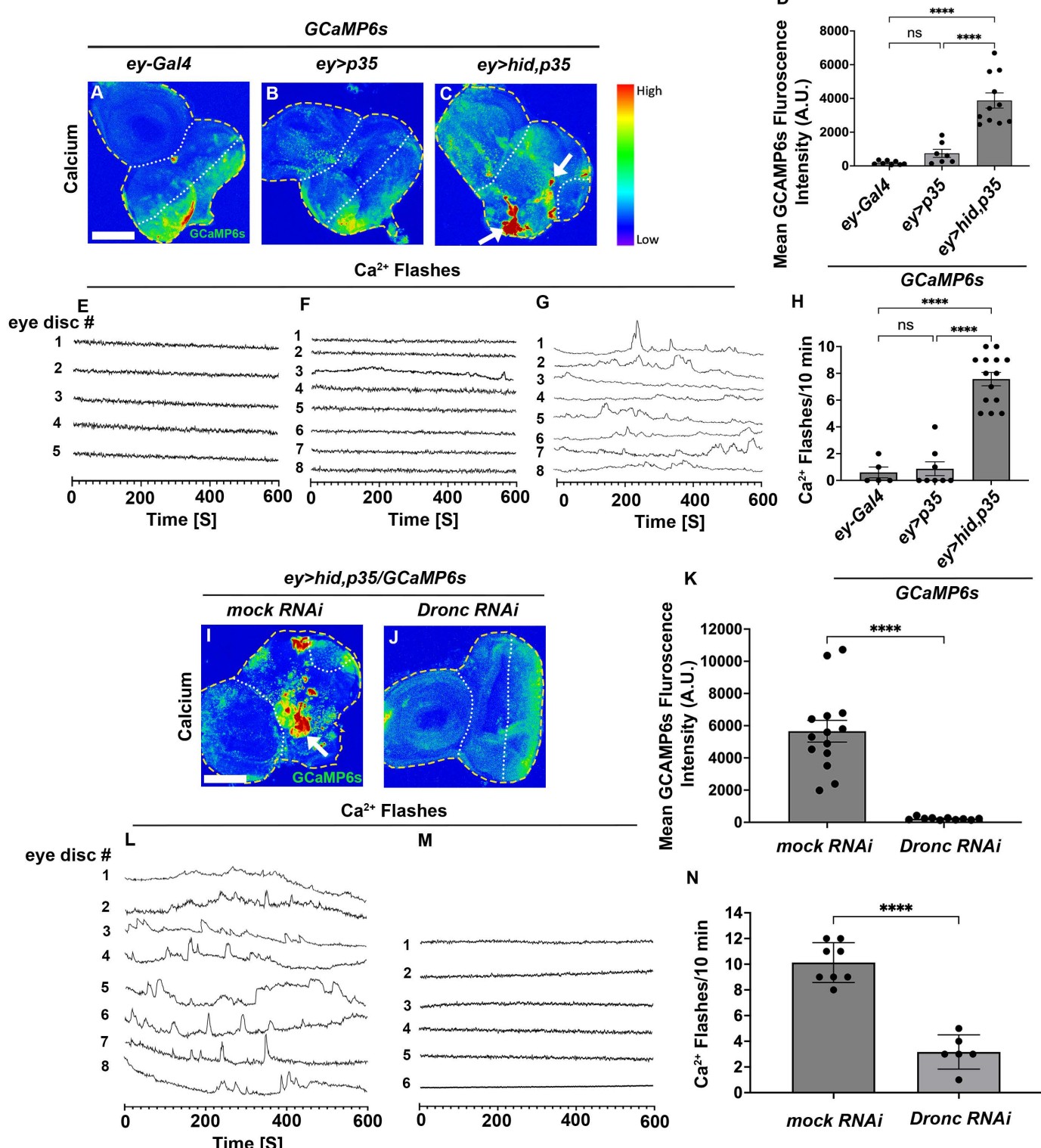

**Fig 2. Undead cells up-regulate cytosolic Ca²⁺ signaling and produce Ca²⁺ flashes in a *Dronc*-dependent manner. (See also S3 Fig).** The disc outline is marked by yellow dashed lines. The white dotted lines highlight the *ey-Gal4*-expressing area of the eye discs. For all panels, scale bars are 50 μm. **(A–C)** Confocal images of third instar larval eye imaginal disc expressing the Ca²⁺ reporter GCaMP6s in control (*ey-Gal4* and *ey>p35*) and in

overgrown, undead (*ey>hid,p35*) discs. The GCaMP6s fluorescence directly represents the cytosolic Ca²⁺ level. The cytosolic Ca²⁺ level is significantly upregulated in undead (*ey>hid,p35*) discs compared to control discs (see white arrows in (**C**)). (**D**) Quantification of cytosolic Ca²⁺ level via measuring *GCaMP6s* fluorescence intensity in *ey-Gal4*-expressing areas of control (*ey-Gal4* and *ey>p35*) eye discs and overgrown, undead (*ey>hid,p35*) eye discs. Data from $n = 8$ (*ey-Gal4*), 7 (*ey>p35*) and 11 (*ey>hid,p35*) discs were analyzed in three independent experiments. A.U—arbitrary units. (**E–G**) Representative Ca²⁺ traces of eye imaginal discs expressing GCaMP6s obtained by time-lapse confocal imaging (600 frames, 1-second intervals). Ca²⁺ flashes refer to transient, localized increases in intracellular calcium levels, detected as brief bursts of elevated GCaMP6s fluorescence. These flashes are characterized by a rapid onset, short duration, and return to baseline, indicating discrete calcium signaling events within imaginal disc cells. Each line represents an independent disc (numbered). For complete individual recordings of the Ca²⁺ traces see S3 Fig. Undead discs display a strong increase in the number of Ca²⁺ flashes (S1–S3 Movies). Genotypes: (E) *ey-Gal4>GCaMP6s* (control) ($n = 5$); (F) *ey-p35>GCaMP6s* ($n = 8$); (G) *ey>hid,p35/GCaMP6s* ($n = 8$). (**H**) Quantification of Ca²⁺ flashes in (**E–G**). Data from $n = 5$ (*ey-Gal4*), 8 (*ey>p35*), and 14 (*ey>hid,p35*) discs were analyzed in three independent experiments. (**I, J**) Confocal images of third instar larval eye imaginal disc expressing the *GCaMP6s* reporter together with *mock* (*Luciferase*) *RNAi* (**I**), and *UAS-Dronc RNAi* (**J**). GCaMP6s fluorescence is indicated by white arrows. (**K**) Quantification of cytosolic Ca²⁺ levels in (**I, J**). Data from $n = 14$ (mock RNAi) and 10 (UAS-Dronc RNAi) discs were analyzed in three independent experiments. A.U.—arbitrary units. (**L, M**) Representative Ca²⁺ traces of eye imaginal discs expressing GCaMP6s obtained by time-lapse confocal imaging (600 frames, 1-second intervals). Each line represents an independent disc (numbered). For complete individual recordings of the Ca²⁺ traces see S3 Fig. The Ca²⁺ flashes are strongly reduced by the expression of UAS-Dronc RNAi (S4 and S5 Movies). Genotypes: (L) *ey>hid,p35/GCaMP6s/Luciferase RNAi* ($n = 8$); (M) *ey>hid,p35/GCaMP6s/ Dronc RNAi* ($n = 6$). (N) Quantification of cytosolic Ca²⁺ flashes in (**L, M**). Data from $n = 8$ (*mock RNAi*) and 6 (*UAS-Dronc* RNAi) discs were analyzed in three independent experiments. The data underlying the graphs shown in this figure can be found in S1 Data.

## Identification and characterization of RyR as intracellular Ca²⁺ channel

CICR from intracellular stores, particularly the ER, serves as an essential mechanism for amplifying calcium signals in many cell types [65,66]. This process is initiated when low levels of Ca²⁺ bind to and activate the RyR at the ER membrane, triggering the release of additional Ca²⁺ from ER stores [83]. Through CICR, cells can rapidly amplify Ca²⁺ signals and extend their reach to distal regions within large cells, including muscle fibers, neurons, and also in epithelial cells. RyR-mediated Ca²⁺ release typically manifests as transient Ca²⁺ flashes [84]. Because we observed a similar Ca²⁺ dynamics in undead cells (S3 and S4 Movies), we investigated the role of the single *Drosophila RyR* gene in Ca²⁺ signaling within undead cells.

*RyR* RNAi moderately suppressed the undead overgrowth of *ey>hid,p35* animals (Fig 4A and 4B). We also observed a significant reduction in GCaMP6s signaling (Fig 4C and 4D; quantified in Fig 4E) as well as Ca²⁺ flashes by *RyR* RNAi in *ey>hid,p35* imaginal discs (Fig 4F and 4G; quantified in Fig 4H and S6 Fig and S4 and S9 Movies).

The reduction in Ca²⁺ signaling by *RyR* RNAi has cascading effects on downstream processes. *RyR* knockdown strongly suppressed ROS generation in undead discs (Fig 4I and 4J; quantified in Fig 4K), consequently preventing hemocyte recruitment to these discs (Fig 4L and 4M; quantified in Fig 4N). Moreover, loss of *RyR* abolished both JNK and Wg signaling, key components of the AiP network (S7 Fig). Collectively, these findings demonstrate that RyR and CICR are essential for proper calcium signaling in undead tissue during AiP.

To further examine the functional roles of these Ca²⁺ channels in undead tissue, we determined whether they are transcriptionally regulated by measuring their mRNA expression using qRT-PCR (Fig 4O). Relative to GAPDH normalization, *TrpM*, *TrpA1*, and *RyR* transcripts were significantly upregulated by 1.7, 1.8, and 3.6 fold, respectively, in undead discs (*ey>hid,p35*) compared to both control conditions (*ey-Gal4* and *ey>p35*) (Fig 4O). Among these, RyR exhibited the strongest induction, consistent with its functional requirement in sustaining AiP-associated signaling. These transcriptional changes support our genetic findings, demonstrating that the upregulation of these Ca²⁺ channels coincides with their critical role in mediating Ca²⁺ influx, DUOX activation, and downstream AiP responses.

### *TrpA1* and *RyR* contribute to regeneration in the *DEts>hid* "genuine" AiP model

Given that TrpM, TrpA1, Pkd2, and RyR are essential for Ca²⁺ influx and DUOX activation in the undead AiP model (Figs 3 and 4), we tested whether these channels are also involved in "genuine" (P35-independent) regeneration using the

 

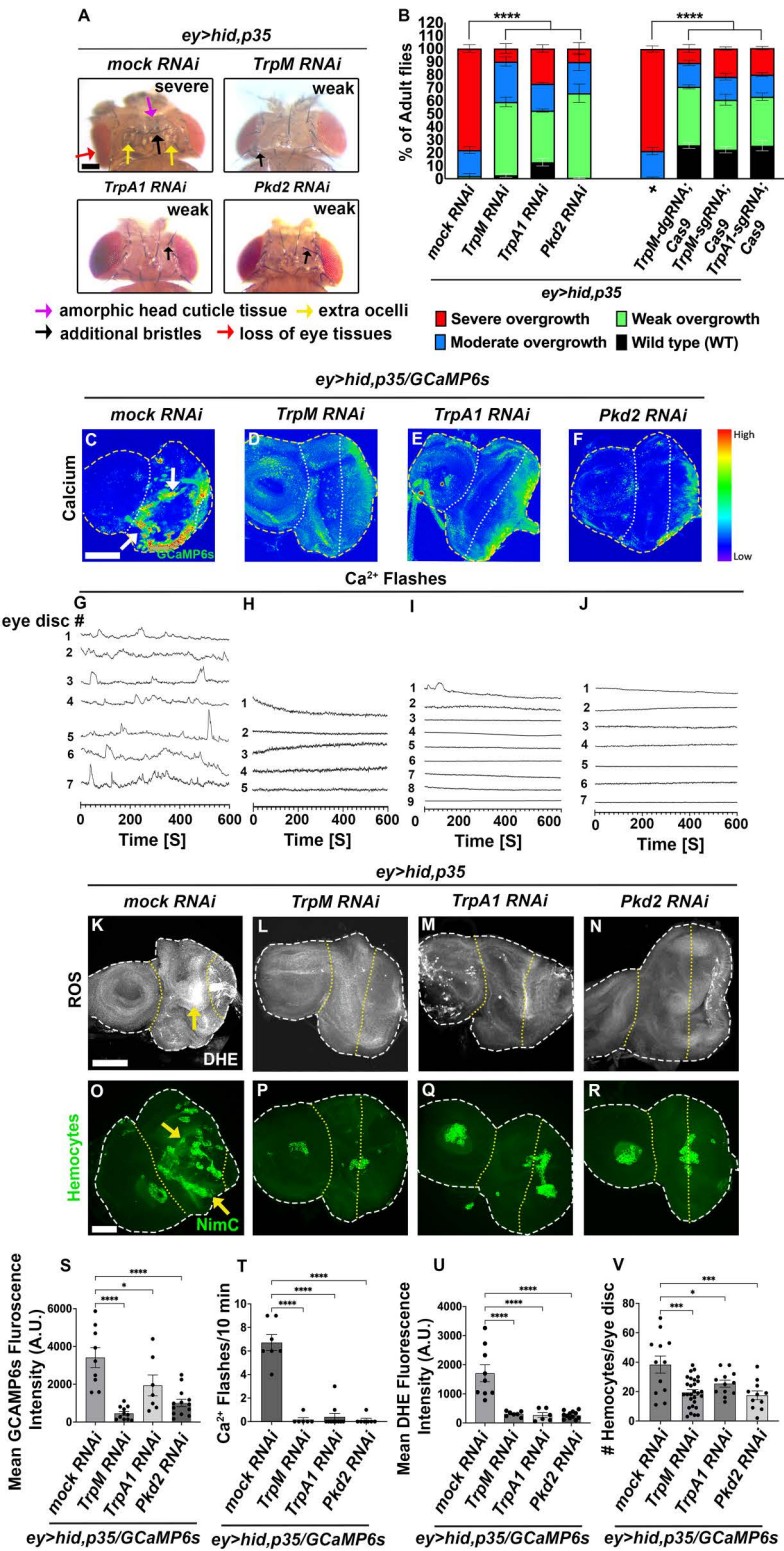

**Fig 3. Three TRP channels are required for cytosolic Ca²⁺ entry and AiP. (See also S4, S5, and S6 Figs).** The disc outlines are marked by white dashed lines. The yellow dotted line highlights the *ey-Gal4*-expressing areas of the eye discs. Scale bars 100 µm (A) and 50 µm (C–F, K–R). **(A)** Representative examples of head overgrowth phenotypes of adult *ey>hid,p35* flies expressing mock (*Luciferase*) RNAi, *UAS-TrpM* RNAi, *UAS-TrpA1* RNAi,

and *UAS-Pkd2* RNAi. Arrows indicate amorphic head capsule tissue (purple arrows), additional bristles (black arrows), and ocelli (yellow arrows) as well as reduced eye tissue (red arrow) in *ey>hid,p35* flies expressing mock (*Luciferase*) RNAi, while black arrows in *UAS-TrpM* RNAi, *UAS-TrpA1* RNAi, and *UAS-Pkd2* RNAi point to one or two extra bristles. Scale bar, 100 μm. **(B)** Quantification of the suppression of the adult *ey>hid,p35*-induced overgrowth phenotype by mock (*Luciferase*) RNAi, *UAS-TrpM* and *UAS-TrpA1* RNAi, and CRISPR/Cas9 inactivation as well as *UAS-Pkd2* RNAi. Progeny was scored as wild type (wt) (black bars), weak (green bars), moderate (blue bars), or severe overgrown (red bars) according to Fig 1A. $n = 100$ flies were counted per genotype in three independent experiments. **(C–F)** Confocal images of third instar larval eye imaginal discs expressing the Ca2+ reporter *GCaMP6s* in *ey>hid,p35* discs expressing mock (*Luciferase*) RNAi, *UAS-TrpM* RNAi, *UAS-TrpA1* RNAi, and *UAS-Pkd2* RNAi. GCaMP6s fluorescence directly corresponds to cytosolic $Ca^{2+}$ levels. Disc boundaries are outlined with yellow dashed lines, and white dotted lines delineate *ey-Gal4*-expressing areas of the eye discs. Scale bars, 50 μm. Quantification shown in **(S)**. **(G–J)** Representative $Ca^{2+}$ traces of eye imaginal discs expressing GCaMP6s obtained by time-lapse confocal imaging (600 frames, 1-second intervals). Each line represents an independent disc (numbered). For complete individual recordings of the $Ca^{2+}$ traces, see S6 Fig. The numbers of $Ca^{2+}$ flashes are strongly reduced by TRP channel RNAi (S6–S8 Movies). Quantification shown in **(T)**. Genotypes: (G) *ey>hid,p35/GCaMP6s/Luciferase* RNAi ($n = 7$); (H) *ey>hid,p35/GCaMP6s/TrpM* RNAi ($n = 5$); (I) *ey>hid,p35/GCaMP6s/TrpA1* RNAi ($n = 9$); (J) *ey>hid,p35/GCaMP6s/Pkd2* RNAi ($n = 7$). **(K–N)** Confocal images of third instar larval eye imaginal *ey>hid,p35* discs expressing mock (*Luciferase*) RNAi, *UAS-TrpM* RNAi, *UAS-TrpA1* RNAi, and *UAS-Pkd2* RNAi. ROS were labeled with dihydroethidium (DHE) dye. Yellow arrows indicate DHE-positive cells. Scale bars, 50 μm. Quantification shown in **(U)**. **(O–R)** Confocal images of hemocytes labeled with the plasmatocyte-specific anti-NimC antibody in undead *ey>hid,p35* discs expressing mock (*Luciferase*) RNAi, *UAS-TrpM* RNAi, *UAS-TrpA1* RNAi, and *UAS-Pkd2* RNAi. Yellow arrows indicate hemocytes. Scale bar, 50 μm. Quantification shown in **(V)**. **(S)** Quantification of cytosolic $Ca^{2+}$ levels shown in (C–F) via measuring *GCaMP6s* fluorescence intensity in *ey>hid,p35* discs expressing mock (*Luciferase*) RNAi, *UAS-TrpM* RNAi, *UAS-TrpA1* RNAi, and *UAS-Pkd2* RNAi. Data from $n = 9$ (*mock* RNAi), 11 (*UAS-TrpM* RNAi), 7 (*UAS-TrpA1* RNAi), and 13 (*UAS-Pkd2* RNAi) discs were analyzed in three independent experiments. A.U.—arbitrary units. **(T)** Quantification of $Ca^{2+}$ flashes shown in (G–J) per 10 min via time-lapse confocal imaging in *ey>hid,p35* discs expressing mock (*Luciferase*) RNAi, *UAS-TrpM* RNAi, *UAS-TrpA1* RNAi, and *UAS-Pkd2* RNAi (S4 and S6–S8 Movies). Data from $n = 7$ (*mock* RNAi), 6 (*UAS-TrpM* RNAi), 10 (*UAS-TrpA1* RNAi), and 7 (*UAS-Pkd2* RNAi) discs were analyzed in three independent experiments. **(U)** Quantification of the DHE fluorescence in **(K–N)**. Data from $n = 9$ (*mock* RNAi), 7 (*UAS-TrpM* RNAi), 6 (*UAS-TrpA1* RNAi), and 12 (*UAS-Pkd2* RNAi) discs were analyzed in three independent experiments. A.U.—arbitrary units. **(V)** Quantification of the number of hemocytes (shown in O–R) in *ey>hid,p35* discs expressing mock (*Luciferase*) RNAi, *UAS-TrpM* RNAi, *UAS-TrpA1* RNAi, and *UAS-Pkd2* RNAi. Data from $n = 12$ (*mock* RNAi), 28 (*UAS-TrpM* RNAi), 12 (*UAS-TrpA1* RNAi), and 11 (*UAS-Pkd2* RNAi) discs were analyzed in three independent experiments. The data underlying the graphs shown in this figure can be found in S1 Data.

*DE^{ts}>hid* model [34]. In this system, *hid* expression is spatially restricted to the dorsal eye disc by *Dorsal Eye-Gal4* (*DE-Gal4*) [85] and temporally controlled by *Gal80^{ts}* [86] through a transient 12-hour temperature shift to 30 °C (Fig 5A). This model co-induces GFP to label *hid*-expressing cells. Compared to controls, 12-hour *hid* expression causes strong apoptosis (Fig 5C) and tissue loss [34]. However, 72 hours post-temperature shift (R72h), discs have fully recovered their shape and normal photoreceptor pattern as evaluated by ELAV labeling (Fig 5D). This recovery results from increased proliferation in the dorsal eye disc [34].

In contrast, RNAi-mediated knockdown of *TrpA1* or *RyR* during the apoptosis-inducing phase impaired regeneration, resulting in only partial recovery of disc structure with defective photoreceptor patterning as revealed by ELAV staining (Fig 5E and 5G; quantified in Fig 5H). The relatively modest effect likely reflects the limited 12-hour RNAi activity during the 30 °C pulse, with channel function restored shortly upon return to 18 °C. Nevertheless, *DE^{ts}>hid* discs in a heterozygous *TrpA1^{ins}* mutant background also showed incomplete regeneration, confirming the requirement for TrpA1 function (Fig 5F; quantified in Fig 5H). Knockdown of these receptors alone using *DE-Gal4* does not affect photoreceptor patterning (S8A–S8D Fig). By contrast, *TrpM* and *Pkd2* knockdown did not affect *DE^{ts}>hid*-induced regeneration (S8E and S8F Fig).

Together, these findings demonstrate that TrpA1 and RyR play significant roles in tissue regeneration following *DE^{ts}>hid*-induced ablation. Furthermore, these data highlight the utility of the simpler "undead" AiP model as a genetic screening platform to identify regulators with broader roles in genuine regenerative responses.

## Discussion

Calcium ($Ca^{2+}$) serves as a dynamic and versatile messenger system in excitable cells, such as neurons and muscles as well as in non-excitable cells, including epithelial cells. In the latter, $Ca^{2+}$ signaling modulates essential processes from stem cell proliferation to immune defence and tissue repair [68,87–89]. Here, we showed that $Ca^{2+}$ signaling also has a

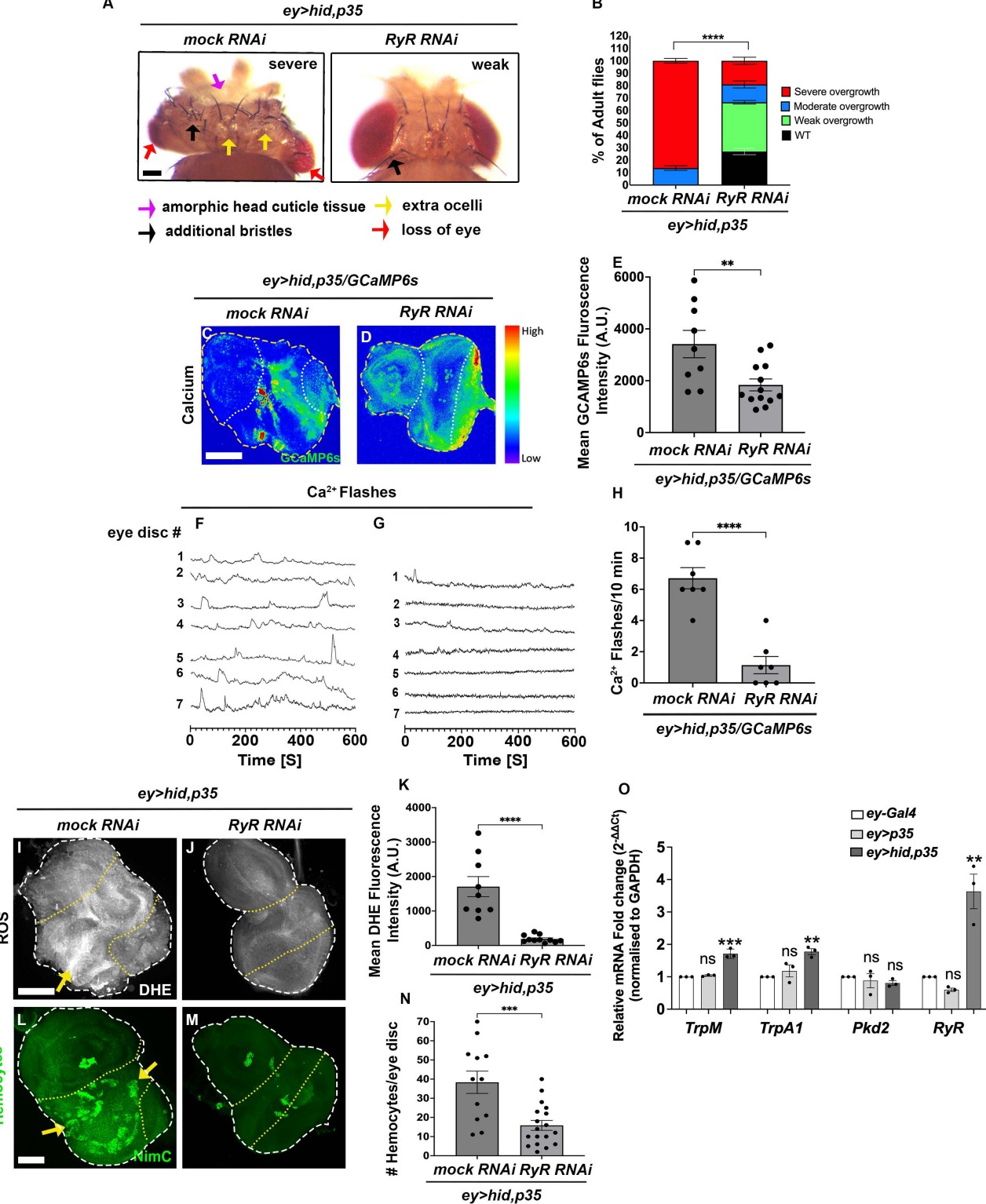

Fig 4. **The Ryanodine Receptor (RyR) is required for cytosolic Ca²+ entry and AiP.** (See also S6 and S7 Fig). The disc outlines are marked by white dashed lines. The yellow dotted line highlights the *ey-Gal4*-expressing area of the eye discs. Scale bars, 100 μm (A), 50 μm (C, D, I, J, L, and M). **(A)** Representative examples of the overgrowth phenotypes of adult *ey>hid,p35* flies head expressing mock (*Luciferase*) RNAi (severe overgrowth) and

UAS-RyR RNAi (weak overgrowth). Arrows in the left panel (ey>hid,p35) point to amorphic tissues (purple arrow), additional bristles (black arrow) and ocelli (yellow arrows), as well as loss of eye tissue (red arrows), while black arrows in the right panel (UAS-RyR RNAi) point to one or two extra bristles, characteristic of weak overgrowth. **(B)** Quantification of the suppression of adult ey>hid,p35-induced overgrowth phenotype by RyR RNAi. Progeny was scored as wild type (WT) (black bars), weak (green bars), moderate (blue bars), or severely overgrown (red bars) according to the classification in Fig 1A. $n = 100$ flies were counted per genotype in three independent experiments. **(C, D)** Confocal imaging of third instar larval eye imaginal disc expressing the $Ca^{2+}$ reporter GCaMP6s in ey>hid,p35 discs expressing mock (Luciferase) RNAi and UAS-RyR RNAi. White arrows in (B) highlight increased $Ca^{2+}$ levels. The disc outline is marked by yellow dashed lines. The white dotted line highlights the ey-Gal4-expressing areas of the eye discs. **(E)** Quantification of the cytosolic $Ca^{2+}$ levels in (C, D) via measuring GCaMP6s fluorescence intensity. Data from $n = 9$ (mock RNAi) and 13 (UAS-RyR RNAi) discs were analyzed in three independent experiments. A.U.—arbitrary units. **(F, G)** Representative $Ca^{2+}$ traces of eye imaginal discs expressing GCaMP6s obtained by time-lapse confocal imaging (600 frames, 1-second intervals). Each line represents an independent disc (numbered). For complete individual recordings of the $Ca^{2+}$ traces, see S6 Fig. The numbers of $Ca^{2+}$ flashes are strongly reduced by RyR RNAi (S4 and S9 Movies). Quantification shown in **(H)**. Genotypes: (F) ey>hid,p35/GCaMP6s/Luciferase RNAi ($n = 7$); (G) ey>hid,p35/GCaMP6s/RyR RNAi ($n = 7$). **(H)** Quantification of $Ca^{2+}$ flashes in **(F, G)**. Data from $n = 7$ (mock RNAi) and 7 (UAS-RyR RNAi) discs were analyzed from three independent experiments. **(I, J)** Confocal images showing third instar larval ey>hid,p35 discs expressing mock (Luciferase) RNAi and UAS-RyR RNAi labeled for ROS with dihydroethidium (DHE) dye. The yellow arrows indicate DHE-positive cells. **(K)** Quantification of the DHE fluorescence levels in **(I, J)**. Data from $n = 9$ (mock RNAi) and 11 (UAS-RyR RNAi) discs were analyzed in three independent experiments. **(L, M)** Confocal images showing hemocytes labeled with the plasmatocyte-specific anti-NimC antibody in third instar larval ey>hid,p35 discs expressing mock (Luciferase) RNAi and UAS-RyR RNAi. Yellow arrows indicate hemocytes. **(N)** Quantification of the number of hemocytes in **(L, M)**. Data from $n = 12$ (mock RNAi) and 18 (UAS-RyR RNAi) discs were analyzed in three independent experiments. **(O)** Relative mRNA levels of TrpM, TrpA1, Pkd2, and RyR in ey-Gal4, ey>p35, and ey>hid,p35 discs measured by qRT-PCR. Data represent the mean of three independent experiments analyzed by one-way ANOVA with Tukey's multiple comparisons test. Error bars represent mean ± SD. The data underlying the graphs shown in this figure can be found in S1 Data.

crucial role in AiP. A Duox mutant lacking essential Glu residues in the EF-hand motifs ($Duox^{EFm}$) displayed reduced ROS levels and suppressed all AiP markers. Because the only known function of the EF-hand motif is $Ca^{2+}$ binding [72,73], it is very likely that they mediate the activation of Duox by $Ca^{2+}$ for the production of ROS during AiP.

Similarly, a previous study on wound repair in Drosophila embryos demonstrated that $Ca^{2+}$ is necessary for DUOX activation and ROS generation [68]. In both cases, wound repair and AiP, the DUOX-generated ROS are needed to recruit hemocytes to wound sites to aid in tissue regeneration [38,68]. Together, these findings illustrate the essential role of $Ca^{2+}$-activated DUOX across cell types and biological processes, highlighting its capacity to drive cellular responses to damage and stress by modulating ROS production and downstream repair mechanisms.

As source(s) of cytosolic $Ca^{2+}$, we identified three specific TRP channels (TrpM, TrpA1, and Pkd2) as well as RyR-mediated CICR which are critical for maintaining $Ca^{2+}$ levels in undead cells, where they are necessary for DUOX activation and ROS generation. Interestingly, the function of these TRP channels in AiP is non-redundant, as inactivating any one channel disrupts $Ca^{2+}$ signaling, blocking ROS production and AiP.

The observation that TrpM, TrpA1, and Pkd2 each appear strictly required for $Ca^{2+}$ signaling in undead discs was unexpected, and suggests that these channels act in a highly interdependent manner. Importantly, TrpM, TrpA1, and Pkd2 belong to different sub-families of the TRP channels with unique structural and functional properties [90], indicating that their requirement may be non-redundant due to unique contributions from each channel sub-family. For example, each channel may respond to distinct inputs generated in undead tissue, such as redox cues, mechanical stress, or metabolic changes, such that all inputs must converge to reach the threshold for CICR and DUOX activation. Another possibility is that they form a functional complex or signaling network in which the activity of each component is required to sustain $Ca^{2+}$ entry. Therefore, disruption of a single channel may destabilize this network, leading to a collapse of $Ca^{2+}$ signaling. Alternatively, the channels may operate in a sequential or feedback-dependent manner, or each may contribute only part of the total $Ca^{2+}$ influx necessary to reach the threshold for DUOX activation. Removing one channel may reduce $Ca^{2+}$ entry below this threshold, which would effectively shut down the cascade. Finally, activation of the RyR could depend on the combined input from all TRP channels to initiate CICR and generate $Ca^{2+}$ flashes.

These possibilities indicate that the TRP channels create a robust but fragile signaling system that collapses when any single component is lost. Such interdependence may ensure that AiP is tightly regulated and triggered only under specific conditions, thereby preventing inappropriate activation.

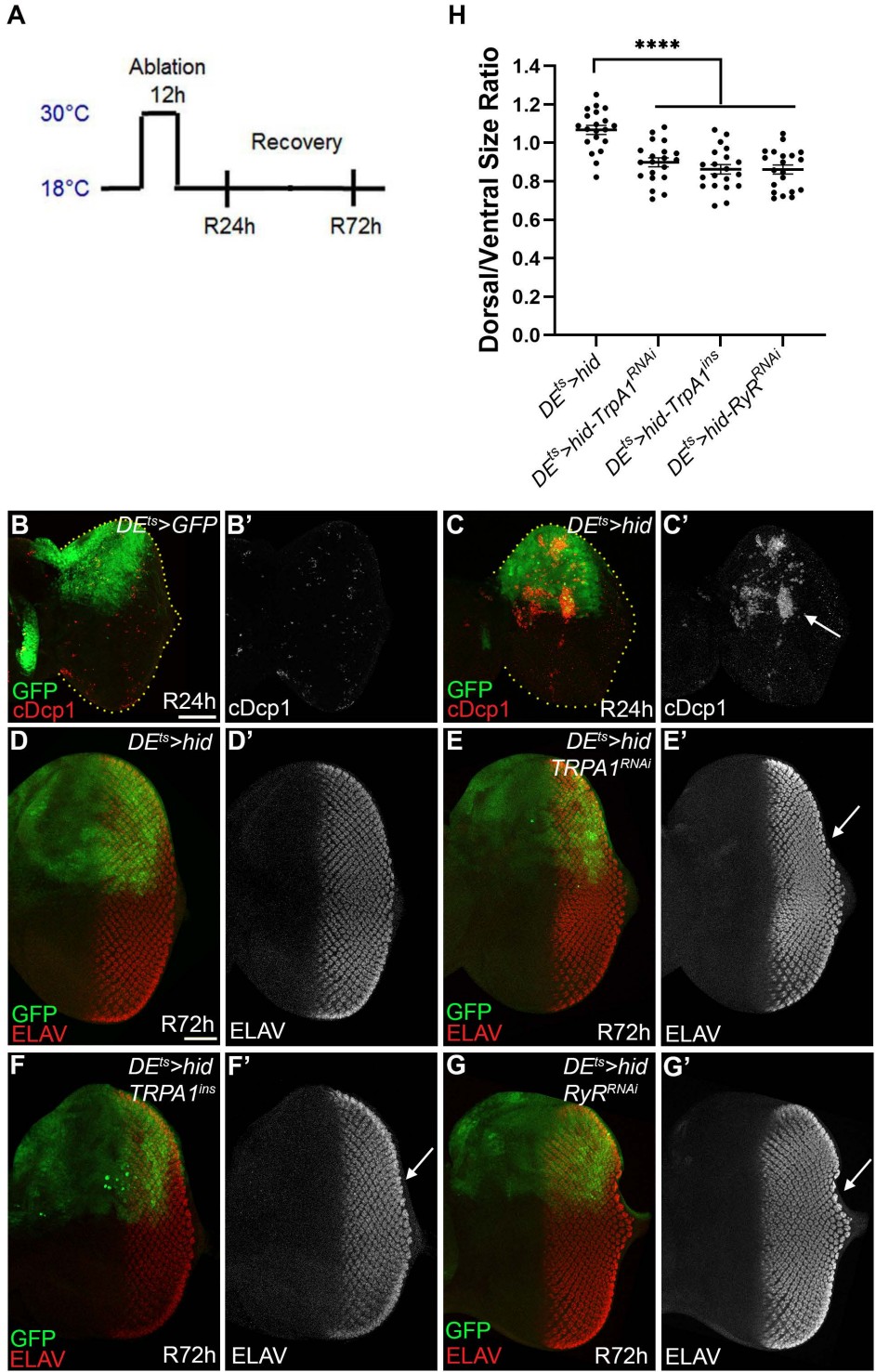

**Fig 5. *TrpA1* and *RyR* are required for complete regeneration in the "genuine" AiP model *DE*ts*>hid*. (See also S8 Fig). (A)** Experimental outline of the ablation/regeneration protocol. Crosses were incubated at 18 °C until 2nd larval instar. Tissue ablation was induced by *hid* expression through temperature shift to 30 °C for 12 hours. Subsequently, larvae were returned to 18 °C and allowed to recover for either 24 h (R24h) or 72 h (R72h) before dissection of imaginal discs. **(B)** Control *DE*ts*>GFP* disc at R24h. Application of the experimental protocol in (A) induces GFP expression (green) in the dorsal half of the eye imaginal disc **(B)**, but does not trigger caspase activation as visualized by cDcp1 labeling (red in B, gray in B'). Scale bar: 100 μm.

**(C)** $DE^{ts}>hid$ disc at R24h. Application of the experimental protocol in (A) triggers extensive caspase activity (cDcp1; red in C, gray in C′) in the dorsal half of the eye imaginal disc (arrow in C′). **(D)** $DE^{ts}>hid$ disc at R72h. The photoreceptor pattern as visualized by ELAV labeling (red in D, gray in D′) develops normally despite strong induction of apoptosis earlier in development according to the protocol in **(A)**. Only 2 out of 20 $DE^{ts}>hid$ discs showed incomplete regeneration. Scale bar: 50 μm. **(E–G)** $DE^{ts}>hid$ discs at R72h expressing $UAS$-$TrpA1$ RNAi (E), TrpA1ins/+ (F), and UAS-RyR RNAi (G). A high percentage of the $DE^{ts}>hid$ discs expressing $UAS$-$TrpA1$ RNAi ($n = 16$ out of 20), $TrpA1^{ins}$ ($n = 17$ out of 20), and $UAS$-$RyR$ RNAi ($n = 15$ out of 20) do not completely recover after 72 h. Arrows in the prime panels highlight incomplete ELAV patterns in the dorsal half of the disc (compare to the ventral half which was not subject to $hid$ expression), indicating that the regeneration response was partially impaired by reduction of $Ca^{2+}$ activity. **(H)** Quantification of dorsal-to-ventral area ratios revealed a significant reduction in experimental genotypes ($TrpA1$ RNAi/$TrpA1^{ins}$/$RyR$ RNAi) compared to controls. Data were analyzed using one-way ANOVA with Tukey's multiple comparisons test in GraphPad Prism. Results are shown as mean ± SEM. ****$p < 0.0001$. The data underlying the graph shown in this panel can be found in S1 Data.

Although the precise mechanism in undead discs remains to be elucidated, similar cases of non-redundant TRP channel function have been described in other systems, and our findings add to this emerging concept. In vertebrates including humans, TRP channels can play similarly specialized non-redundant roles. For example, TrpV4, TrpC7, and several TrpM family members (TrpM2, TrpM4, TrpM8) are upregulated in ovarian cancer, where their expression negatively correlates with patient prognosis [91]. In the trabecular meshwork of the eye, TrpM4 and TrpV4 cooperate to regulate intraocular pressure [92], while in mouse osteoblasts, combined regulation of $Ca^{2+}$ influx by TrpM3 and TrpV4 controls bone remodeling [93]. Together, these examples illustrate that functional division of labor within the TRP family is evolutionarily conserved. The cooperative activation of multiple TRP channels may thus provide both specificity and robustness to $Ca^{2+}$ signaling in complex tissues, ensuring that cells respond appropriately to diverse internal and external cues. [82,90,94–97]

This diversity of TRP function may particularly be beneficial for cells in regenerative contexts, as previously shown in the wound healing in embryos, where TrpM and possibly TrpA1 channels regulate $Ca^{2+}$ influx to control wound repair in response to damage [68]. Thus, TRP channels collectively support dynamic and adaptive $Ca^{2+}$ signaling, essential for both AiP and broader tissue repair mechanisms.

While our data show that Dronc is required for $Ca^{2+}$ release through TRP channels (Fig 2I–2N), the underlying mechanism remains unknown. A direct activation seems unlikely, as caspase-mediated cleavage typically inactivates rather than activates proteins. It therefore remains to be determined whether Dronc acts through its proteolytic activity or via non-proteolytic mechanisms such as scaffolding, binding interactions, or indirect signaling pathways that modulate TRP channel function. Defining this mechanistic link represents an important direction for future research.

CICR, mediated by RyR channels, amplifies $Ca^{2+}$ signaling in cells, a mechanism essential for rapid and sustained cellular responses in vertebrate muscle and neuronal cells [83,98]. We showed here that the RyR channel facilitates CICR in non-excitable cells, amplifying TRP-mediated $Ca^{2+}$ entry to produce $Ca^{2+}$ flashes that drive prolonged AiP signaling. RNAi targeting $RyR$ in undead cells effectively suppresses these $Ca^{2+}$ flashes, disrupting ROS production and hemocyte recruitment, underscoring the essential role of RyR in sustaining $Ca^{2+}$-dependent regenerative signaling.

To compare the roles of the TRP and RyR channels across different contexts of AiP, we also tested their requirement in a genuine AiP model ($DE^{ts}>hid$) (Fig 5). Interestingly, while both the undead and genuine AiP paradigms share a core dependence on $TrpA1$ and $RyR$, the other two TRP channels identified in this study, TrpM and Pkd2, are required only in undead AiP. The additional involvement of TrpM and Pkd2 in undead AiP likely reflects the chronic nature of this system, where sustained caspase activity and prolonged stress signaling necessitate broader channel engagement to maintain elevated cytosolic $Ca^{2+}$ levels. By contrast, genuine AiP represents a transient and self-limiting regenerative response that relies primarily on the TrpA1-RyR axis to generate a short but sufficient $Ca^{2+}$ signal. Despite these differences, the overall mechanism linking $Ca^{2+}$ signaling and undead/regenerative proliferation appears largely conserved between the two AiP models.

Our data collectively support a signaling model involving several key steps (Fig 6). The roles of TRP channels, RyR, and DUOX in AiP and tissue repair highlight a complex but integrated system of $Ca^{2+}$ signaling that supports regenerative

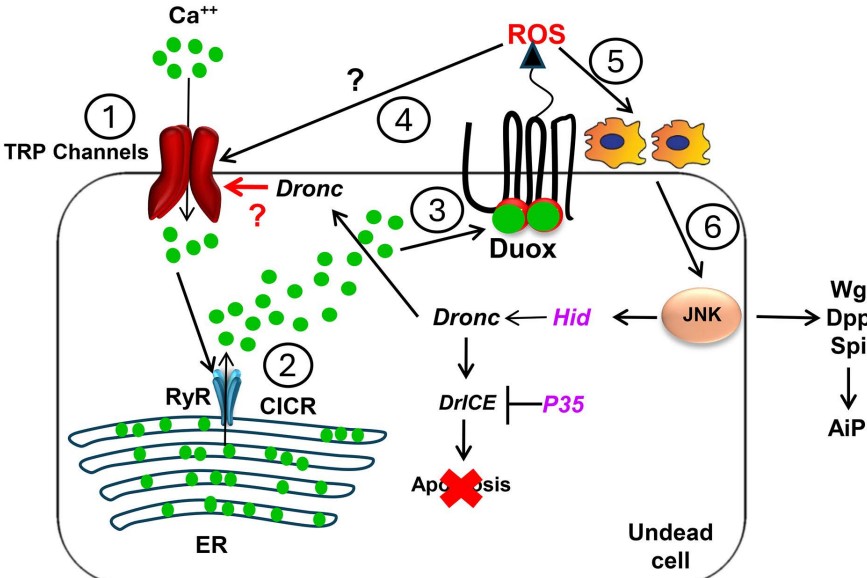

**Fig 6. Summary model.** The proposed mechanism for Ca²⁺ influx and its functional role in apoptosis-induced proliferation (AiP) in undead cells proceeds through multiple steps: (1) Initially, the initiator caspase Dronc is localized to the plasma membrane [39] where it facilitates the activation of TRP channels via a direct or indirect mechanism that remains uncharacterized, enabling Ca²⁺ entry into the cytosol. (2) This primary Ca²⁺ influx triggers a calcium-induced calcium release (CICR) cascade mediated by ryanodine receptors (RyR) in the ER, potentially establishing Ca²⁺ oscillations that amplify the initial Ca²⁺ signal. (3) Upon reaching a specific concentration threshold, cytosolic Ca²⁺ binds to the EF-hand motifs of DUOX, leading to its activation. (4) The reactive oxygen species (ROS) generated by DUOX may create a positive feedback loop through interaction with redox-sensitive TrpA1 channels, thereby maintaining and amplifying Ca²⁺ signaling within undead cells. (5) The primary function of DUOX-derived ROS, however, is to facilitate hemocyte recruitment to the undead disc. (6) Subsequently, these recruited hemocytes secrete signaling molecules, including TNF/Eiger, which activate JNK signaling in undead cells. This activation ultimately results in the release of mitogenic factors Wingless (Wg), Decapentaplegic (Dpp), and Spitz (Spi), which are essential for AiP progression.

processes across diverse biological contexts. In undead cells, DUOX relies on Ca²⁺ binding to produce ROS, which activates downstream pathways essential for AiP. TRP channels ensure sustained calcium influx, with each channel offering a unique response to environmental stimuli, thus providing a non-redundant mechanism for Ca²⁺-dependent signaling. The RyR amplifies these signals via CICR, generating Ca²⁺ flashes that prolong and stabilize the regenerative response.

Apoptosis, AiP, and Ca²⁺ signaling are interconnected processes that play essential roles in development, homeostasis, and regeneration. AiP challenges the traditional view of cell death as a pure destructive process by demonstrating how apoptotic cells can promote tissue regeneration. Ca²⁺ signaling further complicates this picture by acting as both a promoter of apoptosis and a regulator of survival and proliferation. Understanding how these processes are coordinated in *Drosophila* offers valuable insights that are relevant to human health, including cancer biology, regenerative medicine, and neurodegenerative diseases. By understanding the interplay between apoptosis and Ca²⁺ signaling in the context of AiP, we can uncover new strategies for therapeutic interventions that promote regeneration while preventing unwanted cell death.

## Materials and methods

### Fly stocks and genetics

The following transgenic and mutant stocks were used: *ey-Gal4*, *ey>p35* (exact genotype: *ey-Gal4 UAS-p35/ CyO*), *ey>hid,p35* (*UAS-hid; ey-Gal4 UAS-p35/CyO,tub-Gal80*) [34]; *UAS-Duox* RNAi (#44), *UAS-Duox^ΔEF* (gifted by Won-Jae Lee, Seoul National University, South Korea) [89], *TrpA1^ins* (gifted from Dr. Paul A. Garrity, Brandeis

University, USA) [99], *DE-Gal4; tub-Gal80^{ts}* [85]. The following strains were obtained from the Bloomington *Drosophila* Stock Center: *UAS-Luciferase* RNAi (BL#31603), *Duox*-gRNA (BL#77305), *UAS-Cas9.P2* (BL#58986), 20X*UAS-IVS-GCaMP6s* (Chrm II, BL#42746) (Chrm III, BL#42749) [79], *UAS-TrpM* RNAi (BL#35581), *UAS-TrpA1* RNAi (BL#31504), *UAS-Pkd2* RNAi (BL#51502), *UAS-RyR* RNAi (BL#28919), *UAS-Trpl* RNAi (BL#26722), *UAS-Pain* RNAi (BL#61299), *UAS-TRP* RNAi (BL#31650), *UAS-Trpγ* RNAi (BL#31298), *UAS pyx* RNAi (BL#31297), *UAS-wtrw* RNAi (BL#31292), *UAS-Nompc* RNAi (BL#31689), *UAS-iav* RNAi (BL#25865), *UAS-nan* RNAi (BL#31295), *UAS-Trpml* RNAi (BL#602188). The following strain was obtained from the Vienna *Drosophila* Resource Center (VDRC): *UAS-Dronc* RNAi (v100424).

The *Duox^{EFm}* mutant (this study) changes two invariant Glu residues in the EF hand motifs of DUOX to Gln (E879Q and E915Q) which disrupt coordination of $Ca^{2+}$. It was generated by CRISPR/Cas9-mediated homology-directed repair (HDR) by WellGenetics (Taiwan) in the endogenous *Duox* gene. A single guide RNA (gRNA) was designed to target exon 7 of *Duox* near the desired mutation sites. The donor construct consisted of ~1 kb upstream and downstream homology arms flanking two engineered substitutions (E879Q and E915Q), encoded by GAG to CAG and GAA to CAA nucleotide changes, respectively. To facilitate genetic screening, the donor plasmid also carried a PBacDsRed cassette containing a 3xP3-DsRed selection marker, flanked by piggyBac terminal repeats, allowing marker excision by piggyBac transposase. Silent PAM mutations were incorporated into the donor to prevent re-cutting by Cas9. The construct was injected into *w^{1118}* embryos expressing Cas9, and transgenic progeny were identified by DsRed eye fluorescence. Correctly edited alleles were validated by PCR amplification and Sanger sequencing across the targeted region. After marker excision, sequencing confirmed the presence of both E879Q and E915Q mutations in the endogenous *Duox* locus. The *Duox^{EFm}* mutant is homozygous lethal.

Flies were reared on standard cornmeal-molasses medium unless noted otherwise. Crossed flies were transferred into fresh food vials every 2 days. The conditional knockdown of different genes was achieved with the *UAS-GAL4* system [100]. The *ey>hid,p35* stock was crossed to *UAS-luciferase* RNAi as control. For CRISPR/Cas9 crosses, the gRNA and *UAS-Cas9* transgenes were first crossed together, before crossing them with *ey>hid,p35*.

## Fly head phenotype screening and quantification

Adult fly heads were imaged using a ZEISS SteREO Discovery.V8 microscope.

All *ey>hid,p35* animals exhibited head overgrowth phenotypes of varying severity. We classified overgrowth into three categories based on the following criteria: severe cases are characterized by amorphic head cuticle tissue (purple arrows, Fig 1A), numerous additional bristles (up to 15) (black arrows, Fig 1A), extra ocelli (yellow arrows, Fig 1A), and loss of eye tissue (red arrows, Fig 1A). Moderate cases displayed head enlargement with fewer additional bristles (up to 6), duplicated ocelli, and smaller eye size. Weak cases showed mild head enlargement with one to two additional bristles (Fig 1A).

We scored offspring of the RNAi crosses with *ey>hid,p35* for suppression of head capsule overgrowth using the criteria described above. The results of the head phenotype screening are presented as the percentage of adult flies displaying wild-type morphology (black bar), weak (green bar), moderate (blue bar), and severe (red bar) overgrowth (see Fig 1B for example).

## ROS staining

ROS staining was performed using DHE dye following a published protocol by [41]. Briefly, unfixed eye-antennal imaginal discs from third instar larvae were dissected in fresh *Drosophila* Schneider's medium (Thermofisher Scientific #21720024), and incubated in DHE solution (Invitrogen #D23107, final concentration 30 μM) for 5 min. Following staining with DHE, eye discs were washed 3X in 1X PBS and subsequently mounted in Vectashield mounting media. Imaging was done immediately using a Zeiss LSM700 confocal microscope.

## Immunofluorescence labeling

Immunofluorescence labeling of eye imaginal discs was performed by following standard protocols [38]. Briefly, eye-antennal imaginal discs were dissected from third instar larvae in cold 1X PBS, then fixed with 4% paraformaldehyde (PFA) for 30 min at RT, rinsed three times in 1X PBS with 0.3% Triton X-100 for 5 min, blocked with Normal Donkey Serum, and stained with primary antibodies overnight at 4 °C. The following primary antibodies were used: mouse anti-NimC (1:100, P1a,P1b; a kind gift from István Andó) [101]; mouse anti-MMP1 (1:50, 3A6B4), rat anti-ELAV (1:50, 7E8A10), mouse anti- Wg (1:200, 4D4) (all from the Developmental Studies Hybridoma Bank (DSHB)), and rabbit anti-cDcp1 (1:500, Cell Signaling Technology #9578).

After incubation with the primary antibodies, imaginal discs were washed three times in 0.3% PBST and incubated with secondary antibodies and Hoechst 33342 (1:1,000, Cat#3570, Invitrogen) in PBST for 2.5 hr in the dark at RT. Secondary antibodies were anti-mouse IgGs conjugated to Alexa488 and anti-rat IgGs conjugated to Alexa647 (used at 1:20 and 1:30 dilution, respectively, Molecular Probes). Eye discs were counter-labeled with the nuclear dye Hoechst 33342 solution to visualize tissue outline. Discs were then washed 3× in PBST, followed by two times in 1× PBS and mounted in Vectashield.

Images of eye imaginal discs were captured either using a Zeiss LSM700 or a Nikon Eclipse Ti2 confocal microscope. A Z-stack of 20–40 images covering the eye imaginal discs was acquired and shown in maximum intensity projection.

## Imaging cytosolic Ca$^{2+}$ with GCaMP6s reporters

Cytosolic Ca$^{2+}$ in third instar larval eye imaginal disc was monitored *ex vivo* using *UAS-GCaMP6s* as a Ca$^{2+}$ marker [79]. *UAS-GCaMP6s* was crossed into the experimental background for imaging. Eye imaginal discs were dissected and handled in 1× External Saline Solution (ESS), pH 7.2 (1.2M NaCl, 0.04M MgCl$_2$.6H$_2$O, 0.03M KCl, 0.10M NaHCO$_3$, 0.10M Glucose, 0.10M Sucrose, 0.10M Trehalose, 0.05M TES, 0.10M HEPES, 1.5 mM CaCl$_2$) (gifted by Dr. Yang Xiang) at RT, and captured with a Zeiss LSM700 confocal microscope. To quantify eye imaginal discs with high Ca$^{2+}$ levels in *ey-Gal4*-expressing regions, Z-stack images were acquired and shown in Z-max projection (Fig 2A–2C). For the recording of Ca$^{2+}$ flashes, eye imaginal discs were dissected in 1X ESS and immersed in ESS for time lapse video using a Zeiss LSM700 confocal microscope. One layer of eye imaginal disc was recorded every 1s for 10 min resulting in 600 frames.

## Quantification of Ca$^{2+}$ flashes

Ca$^{2+}$ flashes were manually quantified from time-lapse GCaMP6s recordings consisting of 600 frames per disc (10-min duration). For each recording, fluorescence intensity was examined across the entire movie. Ca$^{2+}$ flashes were defined as transient, sharp increases in GCaMP6s fluorescence that was clearly distinguishable from background fluctuations. These events manifest as brief, high-amplitude peaks in intensity traces and short-lived bright spots in the imaging field. Due to variability in spontaneous flash amplitudes across discs, no fixed intensity threshold was applied. Instead, flashes were identified based on morphological criteria: rapid rise in fluorescence, short duration, and return to baseline. Minor fluctuations or slow drifts in signal were excluded. The total number of visually identifiable flashes was recorded for each disc.

## Tissue ablation and recovery using the genuine *DEts>hid* AiP model

For tissue ablation using the genuine *DE$^{ts}$>hid* model, we adapted the protocol developed by [34]. Briefly, larvae of the genotype *UAS-hid/+; UAS-GFP/+; DE-Gal4 tub-Gal80$^{ts}$/+*, either alone (control) or combined with transgenes expressing *TrpA1* and *RyR* RNAi or the *TrpA1$^{ins}$/+* mutant, were raised at 18 °C. Egg laying was allowed for 48h at 18 °C, followed by 5.5 days of larval development at the same temperature and a subsequent 12 h temperature shift to 30 °C to induce *hid* expression. After the heat pulse, larvae were returned to 18 °C for recovery. Imaginal discs were dissected at 24 hours (R24h) or 72 hours (R72h) post-recovery and processed for ELAV or cDcp1 immunolabeling as described above.

## Quantification and statistical analysis

All confocal images were analyzed with Zen3.5 (blue edition) imaging software (Carl Zeiss) and quantified with NIH Fiji software. For quantification of confocal images, the region of interest (*ey-Gal4*-expressing area of eye-antennal imaginal discs) was outlined for each disc and mean fluorescence signal intensity was determined using NIH Fiji software. The $Ca^{2+}$ flashes were quantified using MATLAB (R2022a) software. Data are presented as mean ± SEM from at least three independent experiments. Statistical significance was assessed using one-way ANOVA followed by Dunnett's multiple comparison test (Figs 1B, 3B, and S4A) or a two-tailed unpaired Student *t* test (Figs 4B, S2B, and S5B). For GCaMP6s, DHE, NimC, MMP1, and Wg labelings, fluorescence intensity of the maximum intensity projections was measured. At least three biological repeats were performed for each experiment. Analysis and graph preparation was done using GraphPad Prism 10. Statistical analysis for GCaMP6s, DHE, NimC, MMP1, and Wg fluorescence intensity was performed using one-way ANOVA with Tukey's multiple comparisons test, except for *Dronc*, *RyR* RNAi and *TRPA1^{ins}* experiments, for which statistical analysis was performed using two-tailed unpaired Student *T* test. Data are represented as the mean ± SEM of aggregated data collected from the specified number of samples in each experiment. qRT-PCR results were quantified using one-way ANOVA with Tukey's multiple comparisons test. Graph plotted as Mean ± SD. All figures were assembled with Adobe Photoshop (26.4.1). Levels of significance are depicted by asterisks in the figures: *$p < 0.05$; **$p < 0.01$; ***$p < 0.001$; ****$p < 0.0001$.

## Supporting information

**S1 Fig. Knockdown of *Duox* suppresses JNK activity and *wingless* (*wg*) expression in *ey>hid,p35* discs. (Related to Fig 1).** Disc boundaries are outlined with white dashed lines, and yellow dotted lines delineate *ey-Gal4*-expressing areas of the eye discs. Scale bars represent 50 μm (A–H). **(A–D)** Confocal images of third instar larval eye imaginal discs of control *ey-Gal4* (A, A′), undead (*ey>hid,p35*) discs expressing *mock* (*Luciferase*) RNAi (B, B′), *UAS-Duox* RNAi (C, C′) and *Duox^{EFm}* mutant (D, D′) immunolabeled with MMP1 (a JNK activity marker) and ELAV antibodies. The suppression of the overgrowth phenotype by *UAS-Duox* RNAi and *Duox^{EFm}* (Fig 1) correlates with reduced JNK activity (MMP1; green in A–D; gray in A′, D′; see yellow arrow) and normalization of eye disc patterning as visualized by ELAV labeling (red). **(E–H)** Confocal images of third instar larval eye imaginal discs of control *ey-Gal4* (E, E′), undead (*ey>hid,p35*) discs expressing *mock* (*Luciferase*) RNAi (F, F′), *UAS-Duox* RNAi (G, G′), and *Duox^{EFm}* mutant (H, H′) immunolabeled with Wingless (Wg) and ELAV antibodies. The suppression of the overgrowth phenotype by *UAS-Duox* RNAi and *Duox^{EFm}* (Fig 1) correlates with reduced Wg expression (green in E–H; gray in E′–H′; see yellow arrow) and eye disc patterning was normalized as seen by ELAV labeling (red). **(I)** Quantification of the MMP1 fluorescence levels in (A–D). Data from $n = 11$ (*ey-Gal4*), 14 (*mock* RNAi), 15 (*UAS-Duox* RNAi), and 19 (*Duox^{EFm}*) discs were analyzed in three independent experiments. A.U.—arbitrary units. **(J)** Quantification of the Wg fluorescence levels in (E–H). Data from $n = 10$ (*ey-Gal4*), 16 (*mock* RNAi), 11 (*UAS-Duox* RNAi), and 19 (*Duox^{EFm}*) discs were analyzed in three independent experiments. A.U.—arbitrary units. The data underlying the graphs shown in this figure can be found in S1 Data.
(TIF)

**S2 Fig. Loss of the EF-hands of DUOX suppresses the overgrowth of undead heads and all AiP markers. (Related to Fig 1).** Disc boundaries are outlined with white dashed lines, and yellow dotted lines delineate *ey-Gal4*-expressing areas of the eye discs. Scale bars: 100 μm (B) and 50 μm (C, D, F, G, I, J, K, and L). **(A)** Representative examples of a severely overgrown head of *ey>hid,p35* flies expressing mock (*Luciferase*) RNAi (left) and the suppressed overgrowth of *ey<hid,p35* flies expressing *UAS-Duox* RNAi (right). Arrows point to amorphic tissues (purple arrow), additional bristles (black arrows), and ocelli (yellow arrow), and reduced eye tissue (red arrow). **(B)** Quantification of the suppression of head overgrowth of adult *ey>hid,p35* flies by expression of *UAS-Duox^{ΔEF}*. Progeny was scored as wild type (WT) (black bars), weak (green bars), moderate (blue bars), or severely overgrown (red bars) according to the classification in Fig 1A. $n = 100$ flies counted

per genotype in three independent experiments. **(C, D)** Confocal images of third instar larval *ey>hid,p35* discs expressing *mock* (*Luciferase*) RNAi (C) and *UAS-Duox$^{\Delta EF}$* (D) labeled for ROS with dihydroethidium (DHE) dye. The yellow arrows indicate DHE-positive cells. **(E)** Quantification of the DHE fluorescence levels in (C, D). Data from $n = 11$ (*mock* RNAi) and 18 (*UAS-Duox$^{\Delta EF}$*) discs were analyzed in three independent experiments. A.U.—arbitrary units. **(F, G)** Confocal images showing hemocytes labeled with the plasmatocyte-specific anti-NimC antibody in third instar larval *ey>hid,p35* discs expressing *mock* (*Luciferase*) RNAi (F) and *UAS-Duox$^{\Delta EF}$* (G). Yellow arrows indicate hemocytes. **(H)** Quantification of the number of hemocytes in (F, G). Data from $n = 12$ (*mock* RNAi) and 19 (*UAS-Duox$^{\Delta EF}$*) discs were analyzed in three independent experiments. **(I, J)** Confocal images of third instar larval eye imaginal discs of undead (*ey>hid,p35*) discs expressing *mock* (*Luciferase*) RNAi (I, I′), and *UAS-Duox$^{\Delta EF}$* (J, J′) immunolabeled with MMP1 (a JNK activity marker) and ELAV antibodies. The suppression of the overgrowth phenotype by *UAS-Duox$^{\Delta EF}$* (panels A and B) correlates with reduced JNK activity (MMP1; green in I, J; gray in I′, J′; see yellow arrow) and the normalization of eye disc patterning as seen by ELAV labeling (red). **(K, L)** Confocal images of third instar larval eye imaginal discs of *ey>hid,p35* discs expressing *mock* (*Luciferase*) RNAi (K, K′), and *UAS-Duox$^{\Delta EF}$* (L, L′) immunolabeled with Wg and ELAV antibodies. The suppression of the overgrowth phenotype by *UAS-Duox* RNAi and *UAS-Duox$^{\Delta EF}$* (panels A and B) correlates with reduced Wg expression (green in K, L; gray in K′, L′; see yellow arrow) and the normalization of eye disc patterning as seen by ELAV labeling (red). **(M)** Quantification of the MMP1 fluorescence levels in (I, J). Data from $n = 15$ (*mock* RNAi), and 12 (*UAS-Duox$^{\Delta EF}$*) discs were analyzed in three independent experiments. A.U.—arbitrary units. **(N)** Quantification of the Wg fluorescence levels in (E–H). Data from $n = 15$ (*mock* RNAi) and12 (*UAS-Duox$^{\Delta EF}$*) discs were analyzed in three independent experiments. A.U.—arbitrary units. The data underlying the graphs shown in this figure can be found in S1 Data.
(TIF)

**S3 Fig. Individual time-lapse recordings of Ca²⁺ traces in *GCaMP6s*-expressing eye imaginal discs.** (Related to Fig 2E, 2F, 2G, 2L, and 2M). Shown are the complete time-lapse recordings of the Ca²⁺ traces of the *GCaMP6s*-expressing eye imaginal discs corresponding to those presented in Fig 2E, 2F, 2G, 2L, and 2M. Each trace depicts the fluorescence intensity over time in a single eye disc where peaks in panels (G) and (L) represent the Ca²⁺ flashes occurring in the tissue. Notably, the fluorescence intensity in the undead (*ey>hid,p35*) eye discs is significantly increased compared to controls. Images were acquired for 600 consecutive frames at 1-second intervals (10 min total acquisition time). All recordings were performed under identical imaging conditions to ensure comparability across samples. Genotypes: (A) *ey-Gal4>GCaMP6s* (control) ($n = 5$); (B) *ey-p35>GCaMP6s* ($n = 8$); (C) *ey>hid,p35/GCaMP6s* ($n = 8$); (D) *ey>hid,p35/GCaMP6s/Luciferase RNAi* ($n = 8$); (E) *ey>hid,p35/GCaMP6s/Dronc RNAi* ($n = 8$).
(TIF)

**S4 Fig. Knockdown of three TRP channels suppresses JNK activity and Wg expression. (Related to Fig 3).** **(A)** Summary of the suppression screen targeting all 13 TRP channels in the *D. melanogaster* genome. Quantification of overgrowth suppression of adult *ey>hid,p35* fly heads includes RNAi knockdown of control *mock* (*Luciferase*), *TRP*, *TRPL*, *TRPγ*, *TrpA1*, *Painless* (*Pain*), *Pyrexia* (*pyx*), *Water witch* (*wtwr*), *NompC*, *Inactive* (*iav*), *Nanchung* (*nan*), *TrpM*, *Pkd2*, and *TRPML* genes. Progeny was classified as wild type (wt) (black bars), weak (green bars), moderate (blue bars), or severe overgrown (red bars) based on criteria in Fig 1A. $n = 100$ flies counted per genotype in three independent experiments. *TrpM*, *TrpA1*, and *Pkd2* showed the strongest suppression and were selected for further characterization. Disc boundaries are outlined with white dashed lines, and yellow dotted lines delineate *ey-Gal4*-expressing areas of the eye discs (B–E and F–I). In all panels, Scale bars are 50 µm. **(B–E)** Confocal images of undead third instar larval (*ey>hid,p35*) eye discs expressing *mock* (*Luciferase*) RNAi (B, B′), *UAS-TrpM* RNAi (C, C′), *UAS-TrpA1* RNAi (D, D′), and *UAS-Pkd2* RNAi (E, E′) immunolabeled with MMP1 (JNK marker) and ELAV antibodies. The strong MMP1 labeling in *ey>hid,p35* discs (A, A′; yellow arrow) is strongly suppressed by inactivation of either TRP channel (B–D; B′–D′). ELAV labeling (red) indicates normalization of eye disc patterning upon *TRP* channel knockdown (B–E, red). **(F–I)** Confocal images of undead third instar

larval (*ey>hid,p35*) eye discs expressing *mock* (*Luciferase*) RNAi (F, F′), *UAS-TrpM* RNAi (G, G′), *UAS-TrpA1* RNAi (H, H′), and *UAS-Pkd2* RNAi (I, I′) immunolabeled with Wg (green in F–I; gray in F′–I′; see yellow arrow) and ELAV antibodies (red in F–I). **(J)** Quantification of the MMP1 fluorescence levels in (A–D). Data from $n = 8$ (*mock* RNAi), 12 (*UAS-TrpM* RNAi), 12 (*UAS-TrpA1* RNAi), and 11 (*UAS-Pkd2* RNAi) discs were analyzed in three independent experiments. A.U.—arbitrary units. **(K)** Quantification of the Wg fluorescence levels in (E–H). Data from $n = 14$ (*mock* RNAi), 13 (*UAS-TrpM* RNAi), 12 (*UAS-TrpA1* RNAi), and 15 (*UAS-Pkd2* RNAi) discs were analyzed in three independent experiments. A.U.—arbitrary units. The data underlying the graphs shown in this figure can be found in S1 Data.
(TIF)

**S5 Fig. A *TrpA1* null mutant suppresses overgrowth of undead heads and all AiP markers. (Related to Fig 3).** Disc boundaries are outlined with white dashed lines, and yellow dotted lines delineate *ey-Gal4*-expressing areas of the eye discs (B–E and F–I). Scale bars: 100 μm (A) and 50 μm (C, D, I, J, L, M, O, P, Q, and R). **(A)** Representative examples of a severely overgrown head of *ey>hid,p35* flies expressing mock (*Luciferase*) RNAi and the suppressed overgrowth by *TrpA1ins*/+. Arrows point to amorphic tissues (purple arrow), additional bristles (black arrows) as well as ocelli (yellow arrow) and loss of eye tissue (red arrows) in *ey>hid,p35* expressing mock (*Luciferase*) RNAi, while black arrows in *TrpA1ins*/+ point to one or two extra bristles. **(B)** Quantification of the dominant suppression of head overgrowth of adult *ey>hid,p35* flies by heterozygous *TrpA1ins*/+. Progeny was scored as wild type (wt) (black bars), weak (green bars), moderate (blue bars), or severe overgrowth (red bars) according to the classification in Fig 1A. $n = 100$ flies counted per genotype in three independent experiments. **(C, D)** Confocal images of third instar larval *ey>hid,p35* eye imaginal discs expressing the $Ca^{2+}$ reporter *GCaMP6s* with and without *TrpA1ins*/+. White arrows point to high $Ca^{2+}$ levels. **(E)** Quantification of the cytosolic $Ca^{2+}$ levels in (C, D) via measuring *GCaMP6s* fluorescence intensity. Data from $n = 17$ (*mock RNAi*) and 14 (*TrpA1ins*/+) discs were analyzed in three independent experiments. A.U.—arbitrary units. **(F, G)** Representative $Ca^{2+}$ traces of eye imaginal discs expressing GCaMP6s obtained by time-lapse confocal imaging (600 frames, 1-second intervals). Each line represents an independent disc (numbered). For complete individual recordings of the $Ca^{2+}$ traces, see S6 Fig. The numbers of $Ca^{2+}$ flashes are strongly reduced by *TrpA1ins*/+ (S4 and S10 Movies). Quantification shown in (H). Genotypes: (F) *ey>hid,p35/GCaMP6s/Luciferase* RNAi ($n = 6$); (G) *ey>hid,p35/GCaMP6s/TrpA1ins*/+ ($n = 9$). **(H)** Quantification of $Ca^{2+}$ flashes in (E, F). Data from $n = 7$ (*mock* RNAi) and 11 (*TrpA1ins*/+) discs were analyzed from three independent experiments. **(I, J)** Confocal images of third instar larval *ey>hid,p35* eye imaginal discs with and without *TrpA1ins*/+ labeled for ROS with dihydroethidium (DHE) dye. Yellow arrows indicate DHE-positive cells. **(K)** Quantification of the DHE fluorescence levels in (H, I). Data from $n = 13$ (*mock* RNAi) and 10 (*TrpA1ins*/+) discs were analyzed in three independent experiments. A.U.—arbitrary units. **(L, M)** Confocal images showing hemocytes labeled with the plasmatocyte-specific anti-NimC antibody attached to third instar larval *mock* (*Luciferase*) *RNAi* expressing *ey>hid,p35* discs (K) and in *TrpA1ins*/+ background (L). Yellow arrows indicate hemocytes. **(N)** Quantification of the number of hemocytes per disc in (K,L). Data from $n = 25$ (*mock RNAi*) and 21 (*TrpA1ins*/+) discs were analyzed in three independent experiments. **(O, P)** Confocal images showing third instar larval eye imaginal discs of control undead (*ey>hid,p35*) expressing *mock* (*Luciferase*) RNAi (N, N′) and together with *TrpA1ins*/+ (O, O′), labeled with MMP1 and ELAV antibodies. MMP1 labeling (green in N, O; gray in N′, O′; see yellow arrow) is strongly reduced by *TrpA1ins*/+. ELAV labeling (red) indicates normalization of eye disc patterning. **(Q, R)** Confocal images of third instar larval eye imaginal discs of control undead (*ey>hid,p35*) expressing *mock* (*Luciferase*) RNAi (P, P′) and together with *TrpA1ins*/+ (Q, Q′), immunolabeled with Wg and ELAV antibodies. Wg labeling (green in P, Q; gray in P′, Q′; see yellow arrow) is strongly reduced by *TrpA1ins*/+. ELAV labeling (red) indicates normalization of eye disc patterning. **(S)** Quantification of the MMP1 fluorescence levels in (N, O). Data from $n = 10$ (*mock* RNAi) and 11 (*TrpA1ins*/+) discs were analyzed in three independent experiments. A.U.—arbitrary units. **(T)** Quantification of the Wg fluorescence levels in (P, Q). Data from $n = 10$ (*mock* RNAi) and 11 (*TrpA1ins*/+) discs were analyzed in three independent experiments. The data underlying the graphs shown in this figure can be found in S1 Data.
(TIF)

 

**S6 Fig. Individual time-lapse recordings of Ca²⁺ traces in *GCaMP6s*-expressing eye imaginal discs. (Related to** Figs 3G–3J, 4F, 4G**, and** S5F–S5G**).** Shown are the complete time-lapse recordings of the Ca²⁺ traces of the *GCaMP6s*-expressing eye imaginal discs corresponding to those presented in Figs 3G–3J, 4F, 4G, S4F, and S4G. Each trace depicts the fluorescence intensity over time in a single eye disc where peaks in panels (3G), (4F), and (S5A) represent the Ca²⁺ flashes occurring in the tissue. Notably, the fluorescence intensity in the undead (*ey>hid,p35*) eye discs is significantly increased compared to discs with reduced TRP or RyR function. Images were acquired for 600 consecutive frames at 1-second intervals (10 min total acquisition time). All recordings were performed under identical imaging conditions to ensure comparability across samples. Genotypes: (**A**) *ey>hid,p35/GCaMP6s/Luciferase* RNAi ($n = 7$); (**B**) *ey>hid,p35/GCaMP6s/TrpM* RNAi ($n = 5$); (**C**) *ey>hid,p35/GCaMP6s/TrpA1* RNAi ($n = 9$); (**D**) *ey>hid,p35/GCaMP6s/Pkd2* RNAi ($n = 7$); (**E**) *ey>hid,p35/GCaMP6s/RyR* RNAi ($n = 7$); (**F**) *ey>hid,p35/GCaMP6s/Luciferase* RNAi ($n = 6$); (**G**) *ey>hid,p35/GCaMP6s/TrpA1ⁱⁿˢ/+* ($n = 9$).
(TIF)

**S7 Fig. Loss of *RyR* abolished both JNK and Wg signaling in undead discs. (Related to** Fig 4**).** Disc boundaries are outlined with white dashed lines, and yellow dotted lines delineate *ey-Gal4*-expressing areas of the eye discs (A, B, A′, B′, and C, D, C′, D′). In all panels, Scale bars represent 50μm. (**A, B**) Confocal images of third instar larval eye imaginal discs of undead (*ey>hid,p35*) discs expressing *mock* (*Luciferase*) RNAi (A, A′) and *UAS-RyR* RNAi (B, B′) immunolabeled with MMP1 and ELAV antibodies. MMP1 labeling (green in A, B; gray in A′, B′; see yellow arrow) is strongly reduced by *UAS-RyR* RNAi. ELAV labeling (red) indicates normalization of eye disc patterning by *UAS-RyR* RNAi. (**C, D**) Confocal images showing third instar larval eye imaginal discs of undead (*ey>hid,p35*) discs expressing *mock* (*Luciferase*) *RNAi* (C, C′) and *UAS-RyR* RNAi (D, D″ labeled with Wingless (Wg) and ELAV antibodies. Wg labeling (green in C, D; gray in C′, C′; see yellow arrow) is strongly reduced by *UAS-RyR* RNAi. ELAV labeling (red) indicates normalization of eye disc patterning. (**E**) Quantification of the MMP1 fluorescence levels in (A, B). Data from $n = 8$ (*mock RNAi*) and 13 (*UAS-RyR* RNAi) discs were analyzed in three independent experiments. A.U.—arbitrary units. (**F**) Quantification of the Wg fluorescence levels in (C, D). Data from $n = 14$ (*mock* RNAi), and 11 (*UAS-RyR* RNAi) discs were analyzed in three independent experiments. A.U.—arbitrary units. The data underlying the graphs shown in this figure can be found in S1 Data.
(TIF)

**S8 Fig. *TrpA1* RNAi, *TrpA1*ins/+ ,and *RyR* RNAi do not affect photoreceptor development. (Related to** Fig 5**).** (**A**) Control *DEᵗˢ>GFP* disc at R72h. Following the 12h temperature shift that induces GFP expression (green) in the dorsal half of the eye imaginal discs (see experimental protocol in Fig 5A), photoreceptor patterning appears normal as shown by ELAV labeling (red in A, gray in A′). Scale bar: 50 μm. (**B**) *DEᵗˢ>GFP,TrpA1* RNAi disc at R72h. *TrpA1* RNAi does not affect photoreceptor development following 12h induction at 30 °C and 72h recovery at 18 °C (Fig 5A). ELAV staining appears normal (red in B, gray in B′). GFP expression (green) indicates that transgenes have been induced. (**C**) *DEᵗˢ>GFP*; *TrpA1ⁱⁿˢ* disc at R72h. *TrpA1ⁱⁿˢ/+* does not affect photoreceptor development following 12h incubation at 30 °C and 72h recovery at 18 °C (Fig 5A). ELAV staining appears normal (red in B, gray in B′). (**D**) *DEᵗˢ>GFP,RyR* RNAi disc at R72h. *RyR* RNAi does not affect photoreceptor development following 12h induction at 30 °C and 72h recovery at 18 °C (Fig 5A). ELAV staining appears normal (red in D, gray in D′). GFP expression (green) indicates that transgenes have been induced. (**E, F**) *DEᵗˢ>hid* eye discs at R72h expressing *UAS-TrpM* RNAi (E) and *UAS-Pkd2* RNAi (F) show complete recovery, with all examined discs ($n = 20$ for E; $n = 10$ for F) displaying restored ELAV expression (red in E, F; gray in E′,F′). GFP expression (green) confirms induction of transgenes.
(TIF)

**S1 Data. The data underlying the graphs.**
(XLSX)

**S1 Movie. SM1. *ey-Gal4*.** A total of 10 supplementary movies (SM) were uploaded. Each movie consists of 600 frames in 1-second intervals, i.e., totaling 10 min. The movies are either in MP4 formats and can be watched with Media Player or Elmedia Video Player.
(MP4)

**S2 Movie. SM2. *ey>p35* (*ey-Gal4 UAS-p35*).** A total of 10 supplementary movies (SM) were uploaded. Each movie consists of 600 frames in 1-second intervals, i.e., totaling 10 min. The movies are either in MP4 formats and can be watched with Media Player or Elmedia Video Player.
(MP4)

**S3 Movie. SM3. *ey>hid,p35* (*EHP*).** A total of 10 supplementary movies (SM) were uploaded. Each movie consists of 600 frames in 1-second intervals, i.e., totaling 10 min. The movies are either in MP4 formats and can be watched with Media Player or Elmedia Video Player.
(MP4)

**S4 Movie. SM4. *EHP*+mock (*Luciferase* RNAi).** A total of 10 supplementary movies (SM) were uploaded. Each movie consists of 600 frames in 1-second intervals, i.e., totaling 10 min. The movies are either in MP4 formats and can be watched with Media Player or Elmedia Video Player.
(MP4)

**S5 Movie. SM5. *EHP*+ *UAS-dronc* RNAi.** A total of 10 supplementary movies (SM) were uploaded. Each movie consists of 600 frames in 1-second intervals, i.e., totaling 10 min. The movies are either in MP4 formats and can be watched with Media Player or Elmedia Video Player.
(MP4)

**S6 Movie. SM6. *EHP*+ *UAS-TrpM* RNAi.** A total of 10 supplementary movies (SM) were uploaded. Each movie consists of 600 frames in 1-second intervals, i.e., totaling 10 min. The movies are either in MP4 formats and can be watched with Media Player or Elmedia Video Player.
(MP4)

**S7 Movie. SM7. *EHP*+ *UAS-TrpA1* RNAi.** A total of 10 supplementary movies (SM) were uploaded. Each movie consists of 600 frames in 1-second intervals, i.e., totaling 10 min. The movies are either in MP4 formats and can be watched with Media Player or Elmedia Video Player.
(MP4)

**S8 Movie. SM8. *EHP*+ *UAS-Pkd22* RNAi.** A total of 10 supplementary movies (SM) were uploaded. Each movie consists of 600 frames in 1-second intervals, i.e., totaling 10 min. The movies are either in MP4 formats and can be watched with Media Player or Elmedia Video Player.
(MP4)

**S9 Movie. SM9. *EHP*+ *UAS-RyR* RNAi.** A total of 10 supplementary movies (SM) were uploaded. Each movie consists of 600 frames in 1-second intervals, i.e., totaling 10 min. The movies are either in MP4 formats and can be watched with Media Player or Elmedia Video Player.
(MP4)

**S10 Movie. SM10. *EHP*+ *TrpA1*ins/+.** A total of 10 supplementary movies (SM) were uploaded. Each movie consists of 600 frames in 1-second intervals, i.e., totaling 10 min. The movies are either in MP4 formats and can be watched with Media Player or Elmedia Video Player.
(MP4)

## Acknowledgments

We would like to thank Dr. Fei Wang and Dr. Yang Xiang for their invaluable help with the $Ca^{2+}$ assays; Dr. Paul Garrity, Dr. Won-Jae Lee and Dr. István Andó for fly stocks and antibodies; the Bloomington *Drosophila* Stock Center (supported by NIH grant 5P40OD018537-10) and the Vienna *Drosophila* Resource Center for fly stocks; and the Developmental Studies Hybridoma Bank (DSHB) for antibodies.

## Author contributions

**Conceptualization:** Komal Panchal Suthar, Andreas Bergmann.

**Data curation:** Komal Panchal Suthar, Andreas Bergmann.

**Formal analysis:** Komal Panchal Suthar, Caitlin Hounsell, Yun Fan, Andreas Bergmann.

**Funding acquisition:** Yun Fan, Andreas Bergmann.

**Investigation:** Komal Panchal Suthar, Caitlin Hounsell.

**Methodology:** Komal Panchal Suthar, Caitlin Hounsell.

**Resources:** Andreas Bergmann.

**Supervision:** Andreas Bergmann.

**Validation:** Komal Panchal Suthar, Caitlin Hounsell, Yun Fan.

**Writing – original draft:** Andreas Bergmann.

**Writing – review & editing:** Komal Panchal Suthar, Andreas Bergmann.

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
