## [Editor Report · Decision Letter 0]

7 Mar 2025

Dear Andreas,

Thank you for submitting your manuscript entitled "Role of Calcium Signaling in Apoptosis-induced Proliferation in Drosophila" for consideration as a Short Report by PLOS Biology.

Your manuscript has now been evaluated by the PLOS Biology editorial staff as well as by an academic editor with relevant expertise and I am writing to let you know that we would like to send your submission out for external peer review.

Once your full submission is complete, your paper will undergo a series of checks in preparation for peer review. After your manuscript has passed the checks it will be sent out for review. To provide the metadata for your submission, please Login to Editorial Manager (https://www.editorialmanager.com/pbiology) within two working days, i.e. by Mar 10 2025 11:59PM.

Kind regards,

Ines

--

Ines Alvarez-Garcia, PhD

Senior Editor

PLOS Biology

---

## [Decision Letter · Decision Letter 1]

8 May 2025

Dear Andreas,

Thank you for your patience while your manuscript entitled "Role of Calcium Signaling in Apoptosis-induced Proliferation in Drosophila" was peer-reviewed at PLOS Biology as a Short Report. Please also accept my apologies for the delay in providing you with our decision. Your manuscript has been evaluated by the PLOS Biology editors, an Academic Editor with relevant expertise, and by two independent reviewers.

The reviews are attached below. As you will see, the reviewers find the conclusions potentially interesting, but they also raise several concerns that would need to be addressed before we can consider the manuscript further for publication. Reviewer 1 notes that all the experiments have been performed using the ‘undead’ cell system, but that in this system AiP is prolonged and extended beyond physiological settings, thus the reviewer thinks that experiments should be performed in which apoptosis is induced by a pulse rather than being sustained by ‘undead’ cells; the wild type and TRPA1 mutants should then be compared to confirm the conclusions. Reviewer 2 points out that the full screen has not been included in the paper and it should be added. This reviewer also thinks that it’s unclear how Dronc activates the specific calcium channels and that experiments should be repeated using different controls for different experiments. In addition, the reviewer suggests exploring why there is more GCaMP6s activity in the antennal region of the disc upon generation of undead cells, and quantify Wg and Mmp1 in antennal discs to strengthen the conclusions.

Based on the reviewers’ specific comments and following discussion with the Academic Editor, it is clear that a substantial amount of work would be required to meet the criteria for publication in PLOS Biology. However, given our and the reviewer interest in your study, we would be open to inviting a comprehensive revision of the study that thoroughly addresses all the reviewers' comments. Given the extent of revision that would be needed, we cannot make a decision about publication until we have seen the revised manuscript and your response to the reviewers' comments. Your revised manuscript would need to be seen by the reviewers again, but please note that we would not engage them unless their main concerns have been addressed.

We appreciate that these requests represent a great deal of extra work, and we are willing to relax our standard revision time to allow you 6 months to revise your study. Please email us (plosbiology@plos.org) if you have any questions or concerns, or envision needing a (short) extension.

**IMPORTANT - SUBMITTING YOUR REVISION**

3. Resubmission Checklist

a) *PLOS Data Policy*

b) *Published Peer Review*

Sincerely,

Ines

--

Ines Alvarez-Garcia, PhD

Senior Editor

PLOS Biology

Reviewers' comments

Rev. 1: Paulo Ribeiro - note that this reviewer has signed the review

The report by Suthar and Bergmann addresses the molecular mechanisms regulating the process of apoptosis-induced proliferation (AiP). Building on previous work from the Bergmann laboratory, where several seminal discoveries of the basic biology regulating AiP, the authors now explore the mechanisms leading to the activation of Duox, one of the critical regulators of AiP. The authors show that, in agreement with its protein and domain structure, Duox is activated by Ca2+ and this activation is dependent on Dronc function. The authors also show that the Ca2+ is mobilised from different sources. Ca2+ is internalised from extracellular sources via specific Ca2+ channels and mobilised from the ER to activate Duox. The authors suggest that Ca2+ is therefore essential for AiP.

In general, experiments and data are well presented and appropriately controlled. The results obtained by the authors provide an extra layer of complexity regarding the process of AiP and will be of interest to groups studying apoptosis and regeneration, among others. Comments on the manuscript are shown below.

Major points:

Figure 2: Regarding the analysis of Ca2+ signalling, is there a pattern in the distribution of the Ca2+ flashes? Do they primarily occur in the ey-Gal4-expressing areas or are the surrounding areas also affected?

Related to the point above and given that there is an amplification loop of AiP signalling, is there a role for Duox in the regulation of Ca2+ flashes or AiP-mediated Ca2+ signalling?

Regarding the role of Ca2+ channels, is there an additive or synergistic effect of modulating extracellular Ca2+ channels and RyR regarding AiP phenotypes? Is there a way to test the epistatic relationship between the different Ca2+ channel types (i.e. those controlling extracellular Ca2+ and RyR)?

All the experiments have been performed using the "undead" cell system, which has been exploited at length in the field of AiP. However, by its nature, this is a system where AiP is prolonged and extended beyond what would normally occur in a physiological setting. Given that the authors have a TRPA1 mutant fly stock, would it be possible to conduct experiments where apoptosis is induced in a pulse (i.e. using a FLPout system) rather than sustained using "undead" cells and to compare the extent of AiP in the control situation vs the TRPA1 mutant (where AiP will be affected)? Whilst not strictly necessary, having an experiment conducted in a system beyond the "undead" model would enhance the conclusions of the study.

Related to the point above, authors should comment on the fact that TRPA1 mutants seem to suppress the AiP phenotypes in heterozygosity. Is 50% of gene dose not sufficient to maintain TRPA1 function?

Minor points:

Figure 1A, Figure 3A, Figure S3A, Figure 4A: If possible, statistical analysis of the comparison between the genotypes should be added.

Related to Figure 1, it would be helpful to provide more information regarding the criteria to classify the different phenotypes as weak, moderate or severe overgrowth. This could potentially be included in the relevant Materials and Methods section.

Figure 1C-H, Figure 1I-N and Figure S1: Have the quantifications of the intensity of the different stainings been normalised to tissue size? Moreover, given that AiP affects neighbouring tissue, have the authors analysed and quantified the stainings in the different areas (ey-Gal4-expressing and surrounding areas) separately?

Are both EF hand motifs required? Have the authors created Duox transgenes where only one EF hand has been mutated? If so, do these have intermediate phenotypes?

Related to the action of the Ca2+ channels, is it possible for the authors to specifically inhibit the different channels using chemical inhibitors to further confirm their role and potentially explain the non-redundancy issue? Alternatively, is it possible for authors to over-express (or deplete) Ca2+ regulators with the opposite effect of RyR, TRPA1, TRPM and PKD2 to further validate their results?

Line 215: Typo. S1I should be replaced with S2I.

Line 216: Typo S1J should be replaced with S2J.

Figure 4G: x-axis labels are covered in figure panel.

Line 554: Reference 39 is listed as in press but the correct citation should be provided.

Rev. 2:

The manuscript titled "Role of Calcium Signaling in Apoptosis-induced Proliferation in Drosophila" by Suthar and Bergmann explores the role of cytosolic calcium influx as a crucial event for DUOX activation during apoptosis-induced proliferation (AiP), using the Drosophila eye disc as a model system. This study builds upon previous work demonstrating that activation of the caspase Dronc leads to DUOX activation in "undead" cells, which produces reactive oxygen species (ROS), which recruits immune cells to activate JNK signaling and downstream mitogenic factors that promote proliferation of surrounding cells. In this work, the authors identify the calcium channels - TRPA1, TRPM, PKD2, and RyR - that allow Ca2+ entry into the cytosol when apoptosis is inhibited by p35, activating DUOX via its EF-hand calcium-binding motifs to drive AiP.

To establish the role of these channels, the authors employ a comprehensive suite of genetic tools, including RNA interference, CRISPR-Cas9-mediated knockouts, and mutants. These complementary approaches offer compelling evidence linking calcium channel activity to DUOX activation. However, the authors point out that calcium regulation of DUOX has been previously described, and identification of these channels does not explain how the caspase Dronc activates this pathway. While this manuscript is essentially a genetic screen paper, the screen itself is not presented. Furthermore, the duplication of controls throughout the manuscript raises concerns about the data itself. Thus, substantial work remains before this manuscript is suitable for a journal with a broad readership such as Plos Biology.

Major concerns:

1. Line 192-194: The authors mention a screen was conducted of 13 TRP channel genes. This paper is a genetic screen paper in which the screen itself is not presented. The authors should present the complete screen. How do we know that perturbation of any calcium channel will not have the same effect, and that the ones presented are unique in their role in AiP?

2. The new finding in this paper is identification of the specific calcium channels that act downstream of Dronc to promote Duox activity, as the fact that calcium activates Duox is known. This finding is assuming that presentation of the full screen as requested in point 1 will confirm that the effect on DUOX activity is specific to these four channels. Identification of how Dronc activates these channels would substantially increase the broad applicability of these findings and interest in this work.

3. The controls presented in the figures are identical for each figure - the same disc images are used throughout, sometimes tilted and sometimes not, and the graphs make it clear that the same measurements are graphed for the controls for different experiments. Thus, it appears the controls were done only once, and the numbers re-used for subsequent experiments without running the control side-by-side with the mutant or knockdown being tested. This would render the result invalid, as controls must be performed alongside the experimental condition for immunofluorescence or DHE staining.

a. DHE: 1D, 3N, 4H, S3H, are the same disc. 1H, 3D, 4J, S3J use the same data for the mock RNAi genotype.

b. NimC: 1J, 3R, 4K, S3K are the same disc, slightly rotated. 1N, 3E, S3M use the same data for the mock RNAi genotype.

c. GCamp: 2I, 3F, 4B, S3B are the same disc. 2K and 3B may be the same data plotted slightly differently. 2K, 4D, and S4D are the same data for the mock RNAi genotype.

d. Mmp1: S1B, S2A, S3N, and S4A are the same disc. S1K, S2I, S3R, and S4E are the same data for the mock RNAi genotype.

e. Wg: S1F, S2E, S3P, and S4C are the same disc. S1J, S2J, S3S, and S4S have the same data for the mock RNAi genotype.

Therefore, these experiments should all be repeated with side-by-side controls to confirm the results.

4. In Figure 1B, the four panels depict the extent of the overgrown phenotype, which includes a range of phenotypes relating to bristles, ocelli, eye size etc. The arrows should be changed to different markers pointing at different phenotypes. Importantly, each specific phenotype should be separately quantified to make its prevalence more apparent.

5. Line 199-206: The authors show an interesting finding, that inactivating any single

channel results in loss of GCaMP6s activity, suggesting these channels are non-redundant. However, no experiments were designed to probe the reason behind the fact that all appear necessary and none appear sufficient to activate Duox. The explanation provided in the discussion is unconvincing. If these channels have their own unique inputs, why would inactivating any one channel abolish all or most cytosolic calcium?

6. In the discussion (line 285-287), the authors mentioned some TRP channels were upregulated under certain conditions like cancer. Are they upregulated in the context of AiP as well?

7. In Fig 2C, there seems to be more GCaMP6s activity in the antennal region of the disc upon generation of undead cells, despite the system being restricted to ey+ cells. A similar effect is seen in the RyR RNAi disc (Fig 4C). The authors should explore why this is happening.

8. In Figure S2, Wg and MMP1 also seem to be upregulated in the antennal region, despite the undead cells restricted to the eye disc anterior to the furrow. The authors should quantify Wg and Mmp1 in the antennal discs, and if there is truly elevated expression non-autonomously, explain why.

Minor comments:

1. Line 53, add a period after caspases.

2. In figure 1A, a control for ey>hid,p35 without a mock RNAi should be added, to ensure activating the RNAi machinery does not itself cause any overgrowth.

3. Figure 1A-B: The examples of the phenotypes should be presented before the quantifications.

4. Figure 1: The figure legend stated that the scale bars for C-G are 50 microns and the scale bars for I-M are 100 microns, but the discs in I-M do not look half the size of the discs in C-G. This same concern exists in several other figures.

5. In Figure 1H and N, some bars are incorrectly labeled as "Doux". The labels are too small to read here and in all graphs throughout the other figures.

6. Figure 1H: The legend states that 6 discs were obtained from three experiments. Two discs per experiment is a very low yield - can the authors explain?

7. Figure 2 E-G and other similar panels - does each line represent a different disc? The Y-axis is not labeled (eg Fig2E-G, L etc.)

8. Figure 3 - show the examples of the phenotypes and then show the quantification. Otherwise, the figure panels are called out of order in the text.

9. Figure S3 was called before Figure S2; re-arrangements need to be done accordingly.

10. Figure 4, The images have no scale bars, and it does not seem accurate that K-L are a different scale from H-I.

11. Figure 4G the bottom of the graph is masked or incomplete.

12. Figure S3: The images have no scale bars.

13. Figure S3: the number 1055 is hovering above panel D

14. Videos should all be in MP4 format for universal access

---

## [Decision Letter · Decision Letter 2]

2 Dec 2025

Dear Andreas,

Thank you for your patience while we considered your revised manuscript entitled "Role of Calcium Signaling in Apoptosis-induced Proliferation in Drosophila" for consideration as a Short Report at PLOS Biology. Your revised study has now been evaluated by the PLOS Biology editors, the Academic Editor and the two original reviewers.

In light of the reviews, which you will find at the end of this email, we are pleased to offer you the opportunity to address the remaining points from Reviewer 2 in a revision that we anticipate should not take you very long. We will then assess your revised manuscript and your response to the reviewers' comments with our Academic Editor aiming to avoid further rounds of peer-review.

**IMPORTANT - SUBMITTING YOUR REVISION**

3. Resubmission Checklist

a) *PLOS Data Policy*

Note that we do not require all raw data. Rather, we ask that all individual quantitative observations that underlie the data summarized in the figures and results of your paper be made available in Supplementary files (e.g., excel). Please ensure that all data files are uploaded as 'Supporting Information' and are invariably referred to (in the manuscript, figure legends, and the Description field when uploading your files) using the following format verbatim: S1 Data, S2 Data, etc. Multiple panels of a single or even several figures can be included as multiple sheets in one excel file that is saved using exactly the following convention: S1_Data.xlsx (using an underscore). Please ensure that you provide the individual numerical values that underlie the summary data displayed in the following figure panels as they are essential for readers to assess your analysis and to reproduce it:

Fig. 1B, M, S; Fig. 2D, H, K, N; Fig. 3B, S-V; Fig. 4A, E, H, K, N, O; Fig. S1I, J; Fig. S2B, E, H, M, N; Fig. S4A, J, K; Fig. S5B, E, H, K, N, S, T and Fig. S7E, F

Please also ensure that figure legends in your manuscript include information on WHERE THE UNDERLYING DATA CAN BE FOUND (for example, you can add at the end of the correponding figure legends "The data underlying the graphs shown in the figure can be found in S1 Data"), and ensure your supplemental data file/s has a legend.

For an example see here: http://www.plosbiology.org/article/info%3Adoi%2F10.1371%2Fjournal.pbio.1001908#s5

b) *Published Peer Review*

d) *Code policy*

Sincerely,

Ines

--

Ines Alvarez-Garcia, PhD

Senior Editor

PLOS Biology

Reviewers' comments

Rev. 1: Paulo Ribeiro

The report by Suthar et al. addresses the molecular mechanisms regulating the process of apoptosis-induced proliferation (AiP). Building on previous work from the Bergmann laboratory, where several seminal discoveries of the basic biology regulating AiP, the authors now explore the mechanisms leading to the activation of Duox, one of the critical regulators of AiP. The authors show that, in agreement with its protein and domain structure, Duox is activated by Ca2+ and this activation is dependent on Dronc function. The authors also show that the Ca2+ is mobilised from different sources. Ca2+ is internalised from extracellular sources via specific Ca2+ channels and mobilised from the ER to activate Duox. The authors suggest that Ca2+ is therefore essential for AiP.

The revised manuscript is greatly improved, and the authors strived to address all the major points raised by the reviewers. Importantly, the new data provided enhances and further supports the hypothesis that Ca2+ signalling plays an important role in AiP. The use of a genetic mutant of Duox that is predicted to be unable to bind Ca2+ constitutes an additional genetic system to address the role of Duox and provides additional evidence beyond the use of the dominant-negative version of Duox. The new Duox allele further supports the proposed molecular mechanism presented by the authors. Moreover, the authors also present data in a genetic system that does not result in the long-term accumulation of "undead" cells and their experiments reveal that the mechanism uncovered in this report is likely to be important in more physiological conditions, due to the fact that there was incomplete tissue regeneration when Ca2+ channel function was affected.

Experiments and data are well presented and appropriately controlled. New figure panels are more extensive and provide indication of statistical analyses proposed by the reviewers. Methods include additional information important for clarification of specific points that could be confusing to a more general audience.

In general, the manuscript is vastly improved, includes further evidence supporting the main conclusions of the report and the level of evidence provided is now supported by additional statistical analyses. The data will be of interest to research groups studying apoptosis and regeneration, among others. I support its publication in PLoS Biology.

Rev. 2:

The manuscript titled "Role of Calcium Signaling in Apoptosis-induced Proliferation in Drosophila" by Suthar and Bergmann explores the role of cytosolic calcium influx as a crucial event for DUOX activation during apoptosis-induced proliferation (AiP), using the Drosophila eye disc as a model system. The manuscript is substantially improved and much more complete, with a new way of framing the findings and new data. Some concerns remain, most of which can be addressed without additional experiments.

1. The authors re-used the same control discs in figures 3 and 4 and figures S4 and S7, given that the data are from the same experiment. However, this image re-use gives the impression that there may be only one control disc from each experiment that fits the narrative. Given that the data from these images are quantified, there must be additional control discs that can be used to avoid the re-use.

2. The initial review requested quantification of the individual phenotypes in the adult heads, which the authors declined to do, maintaining their semiquantitative categories of weak, moderate, and strong. It would be less subjective to quantitate the number of extra bristles per head, percent of heads with ectopic ocelli, percent of heads with reduced eyes, etc.

3. Figure 3B and S4A - it is unclear which experiments and controls were conducted at the same time. To be specific, the data for TrpMRNAi, TrpA1RNAi , and Pkd2RNAi in S4A - the original screen - look identical to the data for TrpMRNAi, TrpA1RNAi , and Pdk2RNAi in 3B - the follow-up experiment that tested multiple ways of reducing TrpM and TrpA1 expression. The data for the controls for these two experiments are not identical, but it looks like the data for the three RNAi lines was re-used.

4. Figure 5 E,F, and G - the phenotype should be quantified across sufficient sample size rather than just showing one disc, especially since the phenotype is so subtle. One method of quantification could be the difference in the number of rows of ommatidia between the dorsal and ventral halves of each eye disc.

5. The authors should include a more detailed explanation of the methodology used to quantify the calcium flashes. For example, what constitutes a flash? Signal over a specific magnitude or of a specific duration? This question is raised because in Figure 2H, for the ey>p35 genotype, there is a disc that had 9 flashes and one that had 4 flashes, but none of the traces in 2F/S3B appear to have flashes like those in 2G/S3C, even though the quantification claims that two of them did.

6. It is unclear why the mock RNAi is the appropriate control for the Cas9 knockouts or overexpression of Duox deltaEF and other non-RNAi conditions.

7. The authors use jargon that lacks precision when describing experimental conditions, such as the terms "undead head capsule overgrowth" line 164, "moderately strong suppressed undead overgrowth" line 259, "undead head" line 904, etc.

8. Line 299 describes data not shown - if PLOS Biology has a policy that all data must be shown, these data should be included.

9. For the DuoxEFm mutation, it is unclear if it is heterozygous or homozygous in the experiments. For example, in Line 928 it is written as heterozygous, but is not indicated as heterozygous throughout Figure 1 or S1 labels or line 931, 934, 938, 946, 1131, 1136, 1137, 1141, 1144.

10. Figure 3 Legend A and B are reversed. Similar for S2 Legend A and B.

11. Line 1138 - is Wg activity or expression being monitored?

12. Line1328 should perhaps read 12h temperature shift.

13. Figure 1H-R label each panel with the genotype.

14. Figure 3S-V the notations of statistical significance are too small to see when printed, especially the n.s. in panel V.

15. Figure 4 swapping panels A and B will show the phenotypes before the quantification.

16. Figure 5 A - note the time of egg laying and the time of the temperature shift on the schematic.

---

## [Editor Report · Decision Letter 3]

7 Jan 2026

Dear Dr Bergmann,

Thank you for the submission of your revised Research Article entitled "Calcium signaling regulates apoptosis-induced proliferation in Drosophila" for publication in PLOS Biology. On behalf of my colleagues and the Academic Editor, Nic Tapon, I am delighted to let you know that we can in principle accept your manuscript for publication, provided you address any remaining formatting and reporting issues. These will be detailed in an email you should receive within 2-3 business days from our colleagues in the journal operations team; no action is required from you until then. Please note that we will not be able to formally accept your manuscript and schedule it for publication until you have completed any requested changes.

PRESS

Sincerely,

Ines

--

Ines Alvarez-Garcia, PhD

Senior Editor

PLOS Biology
